# Study on the Extraction Method of Sub-Network for Optimal Operation of Connected and Automated Vehicle-Based Mobility Service and Its Implication

Sehyun Tak [1], Jeongyun Kim [2] and Donghoun Lee [1,*]

1   Center for Connected and Automated Driving Research, The Korea Transport Institute, Sejong 30147, Korea; sehyun.tak@outlook.com
2   Laboratory for Information and Decision Systems, Massachusetts Institute of Technology, Cambridge, MA 02139, USA; kjy@mit.edu
*   Correspondence: donghoun.lee@outlook.com; Tel.: +82-44-211-3311

**Abstract:** There have been enormous efforts to implement automated vehicle-based mobility (AVM) by considering smart infrastructure such as cooperative intelligent transportation system. However, there is lack of consideration on economical approach for an optimal deployment strategy of the AVM service and smart infrastructure. Furthermore, the influence of travel demand in service area has been ignored. We develop a new framework for maximizing the profit of connected and automated vehicle-based mobility (CAV-M) service using cost modeling and metaheuristic optimization algorithm. The proposed framework extracts an optimal sub-network, which is selected by a set of optimal links in the service area, and identifies an optimal construction strategy for the smart infrastructure depending on given operational design domain and travel demand. Based on service network analyses with varying demand patterns and volumes, we observe that the optimal sub-network varies with the combination of trip demand patterns and volumes. It is also found that the benefit of deploying the smart infrastructure is obtainable only when there are sufficient travel demands. Furthermore, the optimal sub-network is always superior to raw network in terms of economical profit, which suggests the proposed framework has great potential to prioritize road links in the target area for the CAV-M service.

**Keywords:** automated vehicle; mobility service; cooperative intelligent transportation systems; travel demand; operational design domain

## 1. Introduction

Automated driving technologies are growing, and it is expected that the influence of autonomous vehicles on traffic networks will continue to increase [1]. Recent advances in autonomous vehicle (AV) technology have attracted considerable attention from both public and private sectors [2]. For obvious reasons, AVs equipped with features, such as automated braking, forward collision warning, and blind spot monitoring, are preferred by most drivers. Besides, AV testing on roadways has been legalized lately in multiple states in the United States. Automated driving is expected to ease traffic-related problems, viz., traffic congestion and accidents [3,4].

However, there are differing views on whether introduction of automated driving would have a positive effect on traffic networks or not [3,5,6]. One optimistic opinion is that automated driving could reduce crashes enormously because 90% of accidents are caused by driver errors [7,8]. Some studies indicate that automated driving could help smoothen traffic flow because it reduces stop-and-go driving, resulting in a reduction in fuel consumption and air pollution [9–14]. More recently, the impact of various factors that affect mixed traffic flow has been actively studied. One of the major factors considered in the literature is the market penetration of AVs, and some studies have pointed out that a

lower mix-ratio of AVs would have a negative effect on the traffic flow, whereas a higher mix-ratio may help improve the traffic flow [15,16]. Factors such as operating dedicated lanes for AVs, the mix-ratio of cooperative AVs, and the level of cautious driving have also been addressed in previous studies [3,17,18]. From the aforementioned literature it is quite obvious that the effect of automated driving varies depending on a combination of these factors.

Just as the impact of automated driving on traffic networks has been revealed from various perspectives, simultaneous studies on the commercialization and adoption of automated mobility services have also been conducted by the automotive industry majors [19]. Especially, companies such as Waymo, Tesla, Apple, Hyundai, and Milo have set up shops to test their vehicles and conducted pilot studies in various cities, such as Silicon Valley, Phoenix, and Arlington. Particularly, Waymo has released whitepapers on the safety methodology for development, testing, deployment, and operations of its autonomous driving technology [20–22]. Through numerous pilot studies, these companies are exploring ways of operating AVs in such a way that it does not initiate conflicts, avoid collisions as far as possible, and mitigate consequences. Nevertheless, one of the biggest challenges for automated driving is arguing that their safety cases are complete [23].

The first step towards a state of informed safety is establishing the capability of an automated driving system (ADS) by defining its operational design domain (ODD) based on safety performance data [21]. The ODD describes the specific conditions under which the ADS is intended to function and refers to the "operating conditions under which a given ADS or feature thereof is specifically designed to function, including, but not limited to, environmental, geographical, time-of-day restrictions, and/or the requisite presence or absence of certain traffic or roadway characteristics" [24]. The ADS is designed to operate based on the operating conditions described in the ODD [25,26]. It is possible to limit the scope of the safety case as well as verification by using the ODD of automated driving to limit the validity of autonomous driving [23].

The problem arises because traditional ODD concepts are combined with mobility service operations, and the areas to which AV-based mobility (AVM) services can be applied based on the restricted ODD are limited: these areas can include the area in which a mobility service is provided which is selected based on customer demand. Moreover, in this case, the appropriate routes for mobility services can also be limited. This inconsistency can be a major barrier for autonomous mobility services because the ODD is restricted to automation levels 1 to 4, as prescribed in the SAE Levels of Driving Automation [24]. It is time for traditional safety-oriented ODD to enter a new phase of expansion for the profitability of mobility services.

To expand the serviceable area of mobility service with automated vehicle restricted by predefined ODD, several studies have focused on securing the driving safety of autonomous vehicles aided by smart infrastructure such as cooperative intelligent transport systems (C-ITS) [27–30]. Several safety-related systems, such as vehicle monitoring systems, road-side alert systems, and collision warning systems, which are still under various stages of development as representative applications of C-ITS have been proposed to accomplish this. The vehicle monitoring system gathers the driving states of vehicles with the collective perception information provided by the C-ITS. The information gathered in the vehicle monitoring system can be used to predict vehicle trajectories as well as evaluate the level of risk from surrounding vehicles [31]. Both road-side alert systems and collision warning systems improve road safety by providing early warning to drivers and help them avoid potential dangers. Connected vehicles equipped with on-board units receive warning signals from roadside units through V2X communication. By enabling the driver to proactively prepare for potentially dangerous situations, C-ITS can prevent accidents as well as mitigate the severity of accidents when they occur [32]. Hence, C-ITS applications related to automated driving that can help expand the service area of automated vehicles have been developed with the goal of improving safety performance. However, from the perspective of providing autonomous driving-based mobility services, there is a general

dearth of studies that answer lingering questions such as—what is the best place to install smart infrastructure?; what kind of smart infrastructure is preferable?; and how effective can they be?

Many previous studies have been conducted on provision of safe and efficient mobility services with automated vehicles. However, these studies on automated vehicles and smart infrastructure such as C-ITS have mainly focused on improving the performance of individual technologies and have failed to take travel demand for mobility services into consideration along with the ODD of automated vehicles and smart infrastructure, even though it is highly related to the service revenue and business model. To improve the safety and profitability of AVM services, it is necessary to understand the influence of factors such as travel demand, ODD of automated vehicles, and smart infrastructure and their combined influence on service revenue. First, the problem of compensating for the mismatch between the restricted ODD and AVM passenger demand needs to be solved by selecting optimal links for service with automated vehicles by considering both ODD and travel demand. Second, the problem related to the limited-service area caused by restricted ODD can be solved by adopting smart infrastructure (e.g., C-ITS) that can improve the safety performance of automated vehicles.

The main objective of this research is to develop a new framework for maximizing the profit of connected and automated vehicle-based mobility (CAV-M) service. The proposed framework selects the optimal links from the road network for the CAV-M service to maximize the profit, where the service area is limited by the ODD. To solve this problem, a new methodology to evaluate the profit of each road link and sub-network is proposed based on graph theory. Additionally, this study proposes an integrated methodology to extract an optimal sub-network for CAV-M services and to establish a contraction strategy for smart infrastructure by using a meta-heuristic algorithm. The remainder of this paper is organized as follows. Section 2 presents the framework of this study and explains the detailed methodology for each analysis. Section 3 deals with the comparison analysis of the raw network and optimal sub-network varying demand pattern and demand amount. Section 4 summarizes the results and suggests directions for future research.

## 2. Methodology

This research proposes sub-network extraction method, which identifies an optimal sub-network with a set of road links in a target area of CAV-M service. The set of road links are determined by link performance evaluation method and optimization with a meta-heuristic algorithm. The following subsections describe the details of the proposed methods. More detailed explanations on network and scenario for case study are also provided as well.

### 2.1. Sub-Network Extraction Method

#### 2.1.1. Link Performance Evaluation Method

This study considers an operational strategy for CAV-M service using a sub-network extraction method. The sub-network extraction method aims to prioritize the introduction of the CAV-M service in a given road network based on profitability assessment. A previous study proposed the strategic optimization of shared autonomous vehicle operations and infrastructure design [33]. Similarly, following the approach of measuring infrastructure cost in the aforementioned reference, the proposed method evaluates the profitability of imposing the CAV-M service with respect to each link involved in the road network. The profitability of introducing the CAV-M service can be measured using (1).

$$C_{Profit} = C_{Revenue}^{Service} + C_{Cost}^{Service} + C_{Benefit}^{Infrastructure} + C_{Cost}^{Infrastructure} \tag{1}$$

where $C_{profit}$ refers to the total profit for imposing the CAV-M service in units of dollars; $C_{Revenue}^{Service}$ and $C_{Cost}^{Service}$ represent the revenue and cost for providing the CAV-M service, respectively, in units of dollars; $C_{Benefit}^{Infrastructure}$ and $C_{Cost}^{Infrastructure}$ indicate the benefit and cost

for installing or/and remodeling, respectively, the existing infrastructure in units of dollars. For instance, to uprate the CAV-M service considering the given ODD of the connected and automated driving, it may either require the installation cost for the smart infrastructure such as C-ITS or the remodeling cost for the current geometric design, such as curvature and grade. With the changes in the existing infrastructure, the CAV-M service can produce some benefits in terms of efficiency and safety.

The first term in $C_{profit}$ is formulated as follows.

$$C_{Revenue}^{Service} = \sum_{i=1}^{n} \left( \frac{C_{Profit\ Margin,\ st}^{Service} \cdot d^i \cdot S_{st}^i}{d_{st}^i} \cdot d_{st}^{i,\ Length\ Effect} \right) \tag{2}$$

where $n$ refers to the number of links in the road network of deploying the CAV-M service, $C_{Profit\ Margin,\ st}^{Service}$ indicates the profit margin per trip considering length of trip, $d^i$ represents the length of link $i$, $S_{st}^i$ denotes the number of trips from node $s$ to node $t$ via link $i$, $d_{st}^i$ describes the trip length of utilizing the shortest path from node $s$ to node $t$ via link $i$, and $d_{st}^{i,\ Length\ Effect}$ is the ratio between the trip length using the shortest path from node $s$ to node $t$ when including and excluding link $i$, as shown in (3).

$$d_{st}^{i,\ Length\ Effect} = \frac{d_{st}^i}{d_{st}^{Excluding}} \tag{3}$$

where $d_{st}^{Excluding}$ represents the trip length when utilizing the shortest path from node $s$ to node $t$. For instance, $C_{Revenue}^{Service}$ tends to decrease as $d_{st}^{Excluding}$ increases when the link $i$ is involved in the set of link sequences for the shortest path.

The second term in $C_{profit}$ is expressed by (4).

$$C_{Cost}^{Service} = -\sum_{i=1}^{n} \left( C_{Cost}^{Operation\ Level} \left( x_{operation}^i \right) \cdot d^i + C_{Cost}^{Safety\ Level} \left( x_{Safety}^{i,\ Pro}, x_{Safety}^{i,\ Sev} \right) \cdot d^i \cdot S_{st}^i \right) \tag{4}$$

where the first and second terms represent the operation and safety costs, respectively, for the CAV-M service; $C_{Cost}^{Operation\ Level}$ is a piecewise cost function of $x_{operation}^i$ to represent the cost per kilometer for applying the CAV-M service from an operational perspective, as shown in (5); $x_{operation}^i$ indicates link $i$'s operational level ranging from 1 to 5 (1: immediately available, 2: map construction, 3: C-ITS such as V2X signal provision, 4: Map construction and C-ITS, 5: Unable to drive); $C_{Cost}^{Safety\ Level}$ is a function of $x_{Safety}^{i,\ Pro}$ and $x_{Safety}^{i,\ Sev}$ to describe the cost per kilometer for each trip to be implemented with the CAV-M service from a safety perspective, as shown in (6); $x_{Safety}^{i,\ Pro}$ and $x_{Safety}^{i,\ Sev}$ represent link $i$'s safety parameters related to collision or/and accident probability and severity, respectively.

$$C_{Cost}^{Operation\ Level} \left( x_{operation}^i \right) = \begin{cases} 1,000,000\ (\$/km), & x_{operation}^i = 5 \\ 100,000\ (\$/km), & x_{operation}^i = 4 \\ 80,000\ (\$/km), & x_{operation}^i = 3 \\ 20,000\ (\$/km), & x_{operation}^i = 2 \\ 1000\ (\$/km), & x_{operation}^i = 1 \end{cases} \tag{5}$$

$$C_{Cost}^{Safety\ Level} \left( x_{Safety}^{i,\ Pro}, x_{Safety}^{i,\ Sev} \right) = P_{Safety}^{Collision} \left( x_{Safety}^{i,\ Pro} \right) \cdot C_{Safety}^{Severity} \left( x_{Safety}^{i,\ Sev} \right) \tag{6}$$

where $P_{Safety}^{Collision} \left( x_{Safety}^{i,\ Pro} \right)$ describes the probability of collision or/and accident in link $i$ considering the geometry and existence of a crosswalk, as shown in (7), and function

$C_{Safety}^{Severity}\left(x_{Safety}^{i,\ Sev}\right)$ represents the severity of a collision or/and accident in link $i$ where collision or/and accident occurs, as shown in (8).

$$P_{Safety}^{Collision}\left(x_{Safety}^{i,\ Pro}\right) = \begin{cases} 0.1\left(\frac{1}{km\cdot trip}\right), & x_{Safety}^{i,\ Pro} = 5 \\ 0.01\left(\frac{1}{km\cdot trip}\right), & x_{Safety}^{i,\ Pro} = 4 \\ 0.001\left(\frac{1}{km\cdot trip}\right), & x_{Safety}^{i,\ Pro} = 3 \\ 0.0001\left(\frac{1}{km\cdot trip}\right), & x_{Safety}^{i,\ Pro} = 2 \\ 0.00001\left(\frac{1}{km\cdot trip}\right), & x_{Safety}^{i,\ Pro} = 1 \end{cases} \qquad (7)$$

$$C_{Safety}^{Severity}\left(x_{Safety}^{i,\ Sev}\right) = \begin{cases} 100,000\ (\$), & x_{Safety}^{i,\ Sev} = 5 \\ 50,000\ (\$), & x_{Safety}^{i,\ Sev} = 4 \\ 10,000\ (\$), & x_{Safety}^{i,\ Sev} = 3 \\ 5000\ (\$), & x_{Safety}^{i,\ Sev} = 2 \\ 1000\ (\$), & x_{Safety}^{i,\ Sev} = 1 \end{cases} \qquad (8)$$

The third term in (1) is formulated as follows:

$$C_{Benefit}^{Infrastructure} = \sum_{i=1}^{n} C_{Gain}^{Infrastructure}\left(a_{Infrastructure}^{i,\ Level}, x_{operation}^{i}, x_{Safety}^{i,\ Pro}, x_{Safety}^{i,\ Sev}, d^{i}, S_{st}^{i}\right), \qquad (9)$$

where $C_{Gain}^{Infrastructrue}$ is a function of $a_{Infrastructure}^{i,\ Level}$, $x_{operation}^{i}$, $x_{Safety}^{i,\ Pro}$, $x_{Safety}^{i,\ Sev}$, $d^{i}$, and $S_{st}^{i}$ to describe the gain by newly constructing the smart infrastructure and/or remodeling the geometric road design and $a_{Infrastructure}^{i,Level}$ represents the actions related to road environment (0: do nothing, 1: constructing the C-ITS infrastructure, 2: remodeling the geometric design of the road, 3: option 1 + option 2). The function $C_{Gain}^{Infrastructrue}$ can be expressed as follows:

$$\begin{aligned} C_{Gain}^{Infrastructure}&\left(a_{Infrastructure}^{i,\ Level}, x_{operation}^{i}, x_{Safety}^{i,\ Pro}, x_{Safety}^{i,\ Sev}, d^{i}, S_{st}^{i}\right) \\ &= G_{Gain}^{Operation}\left(a_{Infrastructure}^{i,\ Level}\right) \cdot C_{Cost}^{Operation\ Level}\left(x_{operation}^{i}\right) \cdot d^{i} + G_{Gain}^{Safety}\left(a_{Infrastructure}^{i,\ Level}\right) \cdot \\ &\quad C_{Cost}^{Safety\ Level}\left(x_{Safety}^{i,\ Pro}, x_{Safety}^{i,\ Sev}\right) \cdot d^{i} \cdot S_{st}^{i}, \end{aligned} \qquad (10)$$

where the first and second terms represent the operation gain and safety gain, respectively, for the changes in infrastructure, $G_{Gain}^{Operation}$ is a piecewise function of $a_{Infrastructure}^{i,Level}$ to represent the ratio of benefit to construction cost from an operational perspective, as shown in (11), and $G_{Gain}^{Safety}$ is a piecewise function of $a_{Infrastructure}^{i,Level}$ to describe the ratio of benefit to construction cost from a safety perspective, as shown in (12).

$$G_{Gain}^{Operation}\left(a_{Infrastructure}^{Level}\right) = \begin{cases} 0.5, & a_{Infrastructure}^{i,Level} = 3 \\ 0.4, & a_{Infrastructure}^{i,Level} = 2 \\ 0.3, & a_{Infrastructure}^{i,Level} = 1 \\ 0, & a_{Infrastructure}^{i,Level} = 0 \end{cases} \qquad (11)$$

$$G_{Gain}^{Safety}\left(a_{Infrastructure}^{Level}\right) = \begin{cases} 0.9, & a_{Infrastructure}^{i,Level} = 3 \\ 0.7, & a_{Infrastructure}^{i,Level} = 2 \\ 0.5, & a_{Infrastructure}^{i,Level} = 1 \\ 0, & a_{Infrastructure}^{i,Level} = 0 \end{cases} \qquad (12)$$

The last term in (1) is formulated as (13):

$$C_{Cost}^{Infrastructure} = -\sum_{i=1}^{n}\left( C_{Cost}^{Installation}\left(a_{Infrastructure}^{i,\ Level}\right) - C_{Cost}^{Current}\left(a_{Infrastructure}^{i,\ Level}\right)\right) \quad (13)$$

where $C_{Cost}^{Installation}$ refers to a function of $a_{Infrastructure}^{i,Level}$ that represents the installation cost for new construction of the smart infrastructure and/or the remodeling cost for the current geometric design of the road, and $C_{Cost}^{Current}$ denotes a function of $a_{Infrastructure}^{i,Level}$ to describe the estimated cost for the existing infrastructure, as shown in (14).

$$C_{Cost}^{Installation}\left(a_{Infrastructure}^{i,\ Level}\right) = C_{Cost}^{Current}\left(a_{Infrastructure}^{i,\ Level}\right) = \begin{cases} 1,000,000\ (\$), & a_{Infrastructure}^{i,Level} = 3 \\ 100,000\ (\$), & a_{Infrastructure}^{i,Level} = 2 \\ 5000\ (\$), & a_{Infrastructure}^{i,Level} = 1 \\ 0\ (\$), & a_{Infrastructure}^{i,Level} = 0 \end{cases} \quad (14)$$

Note that $C_{Cost}^{Current}\left(a_{Infrastructure}^{i,\ Level}\right)$ is the lower bound of $C_{Cost}^{Installation}\left(a_{Infrastructure}^{i,\ Level}\right)$. Thus, $C_{Cost}^{Installation}$ is always less than or equal to zero.

The proposed method allows the service provider to assess the profitability of imposing the CAV-M service with respect to each link in the road network. Therefore, we can find the optimal set of network links to be serviced and the optimal set of network links to construct the smart infrastructure by solving the optimization problem to maximize the total profit for imposing the CAV-M $C_{profit}$.

### 2.1.2. Optimization with Genetic Algorithm

The model presented in this study is a bi-level optimization model. The upper-level problem aims to find the optimal service network for CAV-based services and the lower-level problem aims to find the optimal construction strategy for smart infrastructure. To find the optimal solution for the bi-level optimization model, the problem is cast in the form of a genetic algorithm (GA). The GA was introduced and investigated by [34], which is a representative metaheuristic inspired by the theory of evolution that belongs to evolutionary algorithms. Because GA has diversification strategies while maintaining most of the good-quality solutions, it can be adapted to various problems with high complexity and has been proven to generate high-quality solutions to bi-level and multi-objective optimization problems [35–38]. In the proposed study, the GA is used to find the optimal operation and construction strategy which is an NP-hard problem with multiple decision variables.

A brief description of the implementation of the GA in the proposed model is provided in Figure 1. In the upper-level problem, the GA aims to maximize the sum of the revenue and cost for providing the CAV-M service, i.e., $C_{Revenue}^{Service}$ and $C_{Cost}^{Service}$, with respect to a particular choice of network links to be serviced. The objective variable is a set of network links and all links in the given service network can be included in the optimal set only once. Subsequent to solving the upper-level problem, the optimal set of links becomes the search scope for the solution of the lower-level problem. The lower-level problem finds a set of optimal links to construct a smart infrastructure among the optimal set of network links obtained from the upper-level problem. The GA aims to maximize the total profit by imposing the CAV-M $C_{profit}$ with respect to a particular choice of action for each link.

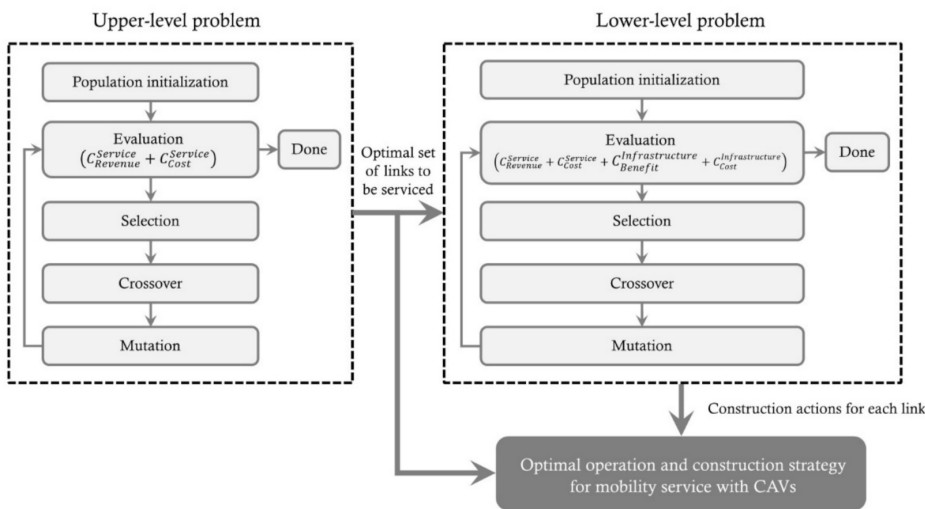

**Figure 1.** Optimization framework to find optimal operation and construction strategy.

### 2.2. Network and Scenario

The proposed operational strategy for the CAV-M service is demonstrated by applying it to a benchmark transportation network case in Sioux Falls, USA, that has been extensively used as a case study in previous studies as well [39–41]. The detailed information on the Sioux Falls network used in this study is described in Appendix A.

To provide an understanding of how the proposed operation and construction strategies for CAV-M work for various trip-demand scenarios, the optimal set of network links for CAV-M service and network links to construct smart infrastructure are evaluated considering six trip demand scenarios, as summarized in Table 1. The six trip-demand scenarios can be grouped under two scenarios. The first scenario group evaluates the influence of the movement patterns of the trip demand: Scenarios 1, 2, and 3 are included in the first group. The second scenario group evaluates the influence of the volume of the trip demand, where the demand is homogeneously distributed to all origin-destination pairs in the network: Scenarios 4, 5, and 6 are included in the second group.

To evaluate the influence of the movement patterns of trip demand, the optimal set of network links to be serviced by the CAV-M and the optimal set of network links that are optimized under Scenarios 1, 2, and 3 are determined first, and then the optimal solutions are compared. As shown in Table 1, the three demand scenarios have different movement patterns while the total number of trips is maintained, as in the original Sioux Falls network (22,857 passengers). Scenario 1 has been created identical to the high demand scenario cited by [39] and consists of movement between external nodes, where the external nodes represent those located at the outermost part of the network. Scenario 2 has been generated as a reverse pattern of Scenario 1 that is composed of the movement from external nodes to internal nodes, where internal nodes represent all except the external nodes. The destination node of each origin-destination pair has been selected randomly among the internal nodes. Scenario 3 is composed of movements between external nodes and from external nodes to internal nodes: the volume of movement of each origin-destination pair has been set to be similar, i.e., between 750 and 1200. Besides, the total volume of movement from each origin node is maintained to be the same as in Scenario 1 and Scenario 2.

The second scenario group evaluated the influence of the volume of the trip demand on the operation and construction strategies. The optimal set of links for mobility service and infrastructure construction is found under three levels of demand: high, medium, and low. Scenarios 4, 5, and 6 correspond to high-, medium-, and low-level scenarios, respectively. These three scenarios were evenly distributed over all 552 combination pairs of nodes. The number of trips per pair is 40, 20, and 19 under Scenario 4, 5, and 6, respectively:

consequently, the total number of movements in Scenarios 4, 5, and 6 are 22,080; 11,040; and 5520, respectively.

**Table 1.** Demand scenarios.

| Origin | Destination | Scenario 1 | Scenario 2 | Scenario 3 | Scenario 4 | Scenario 5 | Scenario 6 |
|---|---|---|---|---|---|---|---|
| S1 | S7 | 2040 | | 1020 | | | |
| S1 | S10 | | 2040 | 1020 | | | |
| S1 | S14 | | 1650 | 825 | | | |
| S1 | S15 | | 2400 | 1200 | | | |
| S1 | S18 | 1650 | | 825 | | | |
| S1 | S20 | 2400 | | 1200 | | | |
| S2 | S6 | 1500 | | 750 | | | |
| S2 | S14 | | 1500 | 750 | | | |
| S2 | S15 | | 1800 | 900 | | | |
| S2 | S18 | 1800 | | 900 | | | |
| S2 | S20 | 1875 | | 937 | | | |
| S2 | S23 | | 1875 | 938 | Evenly Distributed (40 for 552 OD pairs) | Evenly Distributed (20 for each OD) | Evenly Distributed (10 for each OD) |
| S3 | S6 | 2250 | | 1125 | | | |
| S3 | S7 | 1800 | | 900 | | | |
| S3 | S15 | | 2250 | 1125 | | | |
| S3 | S19 | | 1800 | 900 | | | |
| S3 | S20 | 1542 | | 771 | | | |
| S3 | S22 | | 1542 | 771 | | | |
| S13 | S5 | | 2400 | 1200 | | | |
| S13 | S6 | 2400 | | 1200 | | | |
| S13 | S7 | 1500 | | 750 | | | |
| S13 | S8 | | 1500 | 750 | | | |
| S13 | S10 | | 2100 | 1050 | | | |
| S13 | S18 | 2100 | | 1050 | | | |
| | Sum | 22,857 | 22,857 | 22,857 | 22,080 | 11,040 | 5520 |

## 3. Results

This study analyzes the effect of imposing the proposed method on the service area with different demand patterns and sizes. Furthermore, we consider both cases with and without construction of smart infrastructure. The details are provided in the following subsections.

### 3.1. Service Network Analysis Varying Demand Patterns

3.1.1. Optimal Sub-Network without Construction of Smart Infrastructure

Figure 2 shows the result of deriving the optimal subnetwork in Scenarios 1, 2, and 3 when the demand pattern is changed, and the different optimal subnetworks that are derived depending on the demand pattern can be observed.

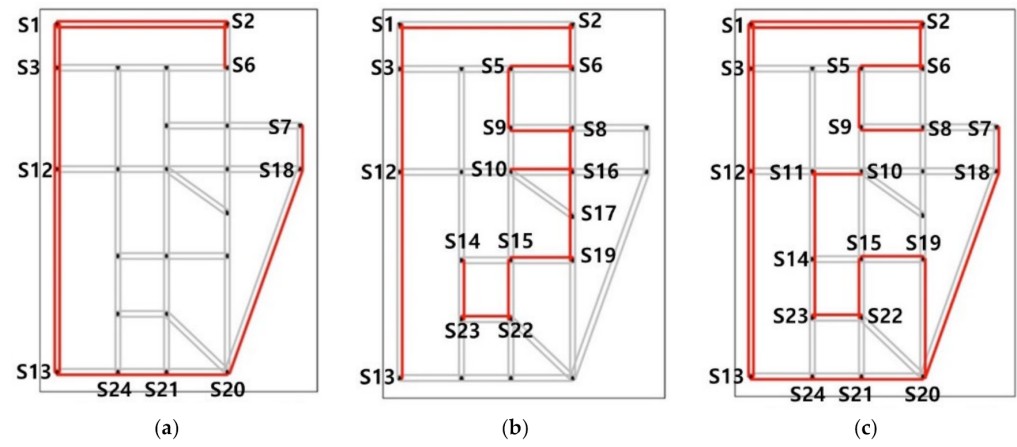

**Figure 2.** Optimal sub-networks for CAV-M service with varying demand patterns: (**a**) Scenario 1, (**b**) Scenario 2, and (**c**) Scenario 3.

Figure 2a shows the result of Scenario 1, where the traffic demand moves from the S1, S2, S3, and S13 nodes located in the outer part of the network to S6, S7, S18, and S20 nodes, which are also located in the outer part of the network. In Scenario 1, where the origin and destination of the traffic demand are located in the outer part, the sub-network for mobility service was derived around the links located in the outer part. Specifically, the use of links for movement is minimized by moving around the outer part rather than using the inner links in the raw network. Figure 2b shows the result of Scenario 2, where the traffic demand moves from the S1, S2, S3, and S13 nodes in the outer part of the network to the S5, S8, S10, S14, S15, S19, S22, and S23 nodes in the inner part of the network. Unlike Scenario 1, in Scenario 2 both the origin and destination of the demand are located in the inner and outer parts, and hence the inner links are actively used in the CAV-based mobility service. Scenario 2 uses both the outer and inner links of the raw network to move the demand in the outer part to the inner part in a cost-effective manner. As shown in Figure 2c, because Scenario 3 is a mix of the demand patterns of Scenarios 1 and 2, an optimal sub-network that combines the optimal sub-networks of Scenarios 1 and 2 has been derived. It can also be observed that a few links that are not used in the optimal sub-networks of Scenarios 1 and 2 are used in the optimal sub-network of Scenario 3. This is because, as different demand patterns are combined, the optimal result is derived for the system if a bypass is selected. Furthermore, to support demand patterns that vary, the number of links used to provide the CAV-M service gradually increased from 13 in Scenario 1, to 16 in Scenario 2, and 24 in Scenario 3.

Figure 3 depicts the demand-weighted betweenness for each demand scenario when the CAV-M service is provided based on the optimal sub-network derived in Figure 2. Betweenness refers to edge betweenness centrality, which is defined as the number of shortest paths that pass through an edge in a graph or network [42]. It is a representative index that indicates the importance of a link which is the number of times that a link is used when the given origin-destination is moved through the shortest path. A link with a high betweenness score indicates that the removal of the link may affect the communication between many pairs of nodes through the shortest paths between them. Betweenness is an index that indicates the relative importance of each link from the network perspective and is used in various research fields, including transportation [43–45]. However, conventional betweenness is limited in that it does not reflect the real magnitude of traffic demand. For example, Scenario 1 in Table 1 has 12 origin-destination pairs and the maximum value of betweenness in this case is 12. More specifically, the traffic demand of S2 (origin)—S6 (destination) with 1500 people moving is treated with the same criticality as the traffic demand of S1 (origin)—S20 (destination) with 2400 people moving. Therefore, a value reflecting the magnitude of a demand is required to consider all the relevant aspects of service provision and profitability. This study analyzes the changes in important links in a network by calculating the demand-weighted betweenness which reflects the magnitude of the demand.

Figure 3 demonstrates that the relative importance of links changes within the optimal sub-network according to the demand pattern. For instance, in Scenario 1 (Figure 3a) 16,000–19,000 people travel through the links S13｜S24, S24｜S21, and S21｜S20 in the bottom outer part of the network, whereas in Scenario 2 (Figure 3b) more than 19,000 people travel through the links S2｜S6, S6｜S5, S5｜S9, and S9｜S8 in the upper right part of the network. On the other hand, in Scenario 3 where the demand patterns of Scenarios 1 and 2 are mixed, as shown in Figure 3c, the links S13｜S24, S24｜S21, and S21｜S20 located in the bottom outer part of the network play an important role. Moreover, unlike Scenarios 1 and 2, the links S20｜S19 and S19｜S15 are used to support the movement of demand towards the inner part of the network. This is considered to be the second-best plan for provision of CAV-M service using links with a high difficulty in applying autonomous driving (e.g., S8｜S16).

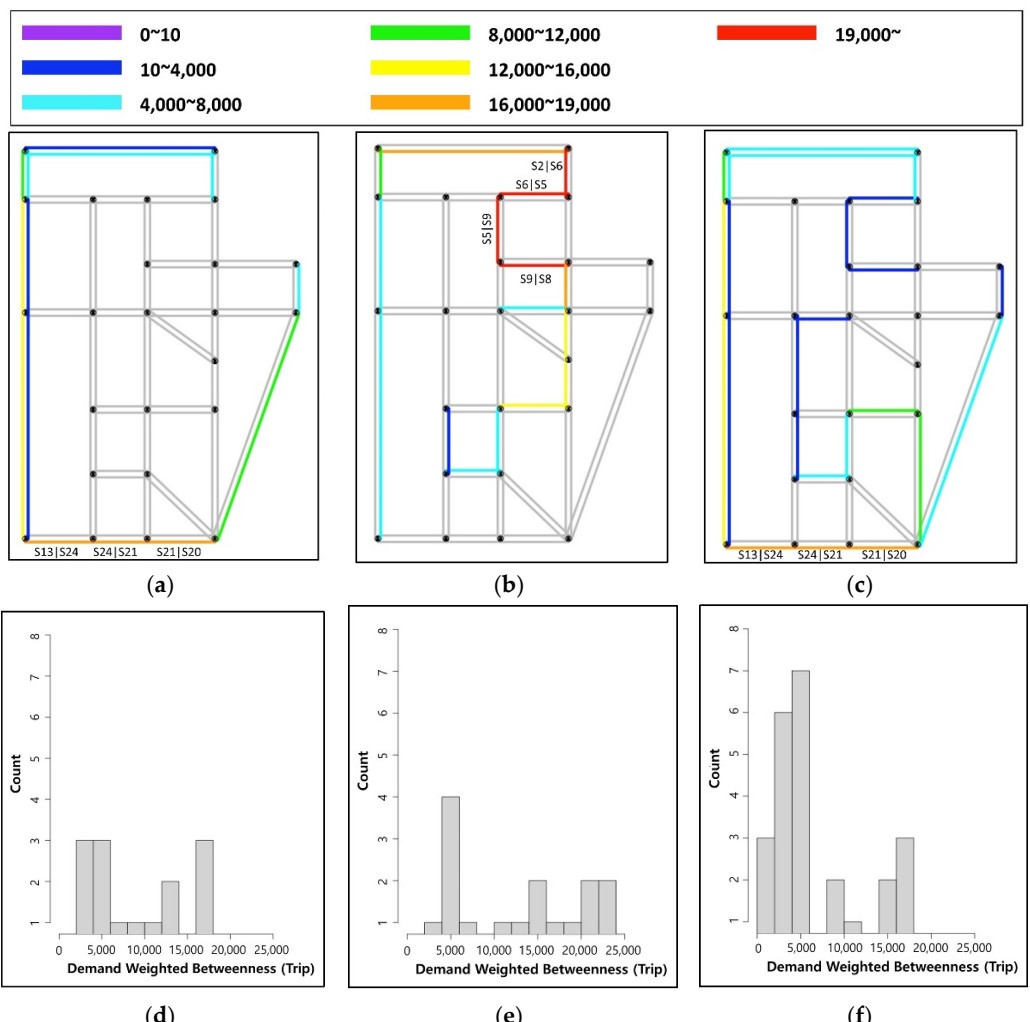

**Figure 3.** Distributions of demand weighted betweenness in Scenario 1, Scenario 2, and Scenario 3. (**a**) demand weighted betweenness in Scenario 1; (**b**) demand weighted betweenness in Scenario 2; (**c**) demand weighted betweenness in Scenario 3; (**d**) distribution of demand weighted betweenness in Scenario 1; (**e**) distribution of demand weighted betweenness in Scenario 2; (**f**) distribution of demand weighted betweenness in Scenario 3.

The change in the relative importance of the network links in correspondence with change in the demand pattern can be seen in Figure 3d–f. In Scenario 1, trips are generated using the links in the outer part of the network and the maximum value of the demand-weighted betweenness score for each link is low at 18,000, whereas the concentration in a specific link is higher in Scenario 2 than that in Scenario 1. Consequently, links that must be treated as important, from the service perspective, are clearly distinguished. Although Scenario 3 has a tendency for the demand to be concentrated in a few links, in comparison to Scenarios 1 and 2, it can be observed that the demand is quite evenly distributed.

Figure 4 shows the results of a few major indices for Scenarios 1, 2, and 3 in the optimal sub-network without considering smart infrastructure, which include (a) service revenue, (b) increasing ratio of shortest path length, (c) operation and safety cost, and (d) profit. As shown in Figure 4a, Scenarios 1, 2, and 3 have the same CAV-M service demands, and because all the demands have been accommodated, the three scenarios show identical service revenue based on Equation (2).

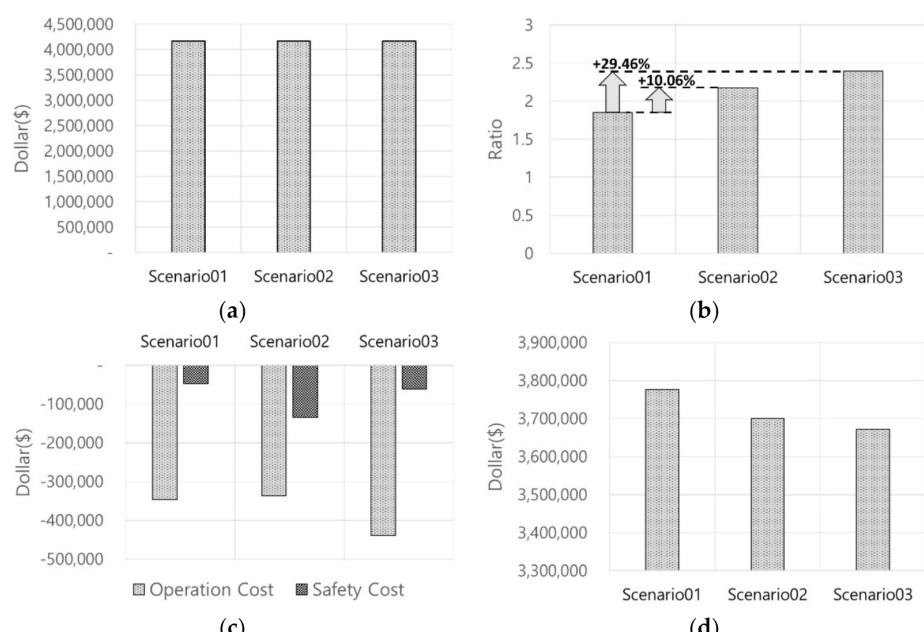

**Figure 4.** (**a**) service revenue, (**b**) increasing ratio of shortest path length, (**c**) operation and safety cost, and (**d**) profit in optimal sub-network without consideration of construction of smart infrastructure.

Furthermore, Figure 4b shows the differences in the increasing ratio of shortest path length, which represents the ratio of the origin-destination (OD) shortest path length in the optimal sub-network and the OD shortest path length in the raw network, among the scenarios. It can be seen that the increasing ratio of the shortest path length is the lowest in Scenario 1 and highest in Scenario 3. In Scenario 2, the increasing ratio of the shortest path length increased by approximately 10% in comparison to Scenario 1, whereas in Scenario 3 the increasing ratio of the shortest path length increased by approximately 29% in comparison to Scenario 1. Specifically, Scenario 1 has the lowest number of bypasses to meet the CAV-M service demand while Scenario 3 has relatively numerous bypasses, and this can be intuitively interpreted in correspondence with the results shown in Figure 3a–c. In contrast, Scenario 3 has a higher number of links used in the network to accommodate the increasing number of OD pairs in comparison to Scenario 1. Consequently, it was set to bypass a large number of trips; hence, it can be seen that the increasing ratio of the shortest path length is clearly different.

Figure 4c shows the result of the operation and safety cost calculated using Equation (4). While there is no significant difference in the operation cost between Scenarios 1 and 2, Scenario 3 requires a higher operation cost for the CAV-M service, whereas Scenario 2 requires a higher safety cost than Scenarios 1 and 3. This is interpreted as the result of providing the CAV-M service through a relatively high-risk path in terms of safety. In contrast, it seems beneficial for Scenario 3, in terms of profitability, to provide CAV-M service by guiding it to a safer route, despite the wide spatial distribution of links used on the network due to the increase of OD pairs.

Lastly, Figure 4d establishes that Scenario 1 has the highest profit followed by Scenarios 2 and 3. A comparison of Scenarios 1 and 2 confirms that the resulting profit varies in correspondence with the difference in safety costs. Scenarios 2 and 3 show similar profits because the difference between the operation and safety costs gets compensated suggesting that a higher profit is possible only if the difficulty of applying autonomous driving on the link to which the service is applied is considered together with the number of links used in the network for provision of CAV-M service.

Figure 5 shows the result of comparison between the raw network and optimal sub-network profit when the smart infrastructure is not considered. Figure 5a shows the differences in the service revenue, operation, and safety costs between the raw network and

the optimal sub-network for each scenario. As shown in the results, there is no significant difference in the service revenue because every scenario has the same CAV-M service demand and the service demand is fully accommodated, but there is a significant difference in the operation and safety costs. Particularly, the outstanding effect of the optimal sub-network for all demand patterns of every scenario can be seen with regard to the safety cost. This suggests that when introducing the CAV-M service while taking into consideration the difficulty of applying autonomous driving, there is a road section that should be provided first because considerable economic benefits can be achieved there.

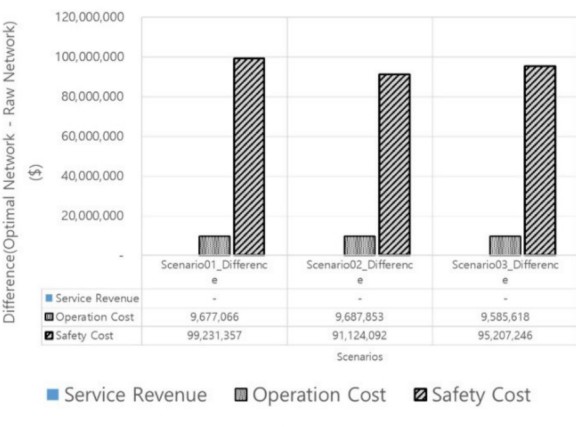

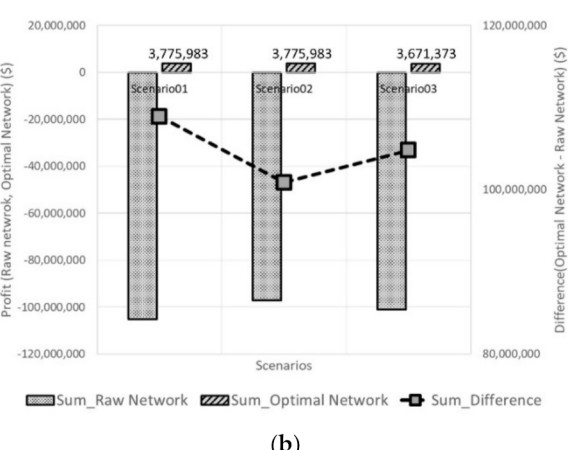

(**a**)　　　　　　　　　　　　　　　　　　(**b**)

**Figure 5.** Comparison analysis of raw network, and optimal sub-network without consideration of construction of smart infrastructure. (**a**) comparison of service revenue, operation cost and safety cost between raw network and optimal network; (**b**) comparison of profit between raw network and optimal network.

Figure 5b shows the result of comparing profit between the raw network and optimal sub-network without considering smart infrastructure. It can be seen that the raw network shows a huge expenditure of approximately $10M, whereas the optimal sub-network achieves a profit of approximately $3.8M. The operation of the CAV-M service in the raw network appears to act favorably for demand patterns corresponding to the order of Scenarios 2, 3, and 1, whereas in the optimal sub-network the advantage in service revenue could be high in the order of Scenarios 1, 2, and 3. With respect to the expected effect of introducing the CAV-M service through the application of the optimal sub-network methodology, the highest utility value appeared in Scenario 1, followed by Scenarios 3 and 2. Therefore, it is expected that various ripple effects would be obtained depending on the spatial demand patterns even though the traffic demand is the same.

### 3.1.2. Optimal Sub-Network with Construction of Smart Infrastructure

Table 2 shows the construction strategy for the smart infrastructure of the optimal sub-network for Scenarios 1, 2, and 3 derived based on GA optimization introduced in Section 2. For example, in the case of the S1 | S2 link of Scenario 1, the smart infrastructure optimization in terms of profitability in consideration of the traffic demand, road length, and difficulty in applying autonomous driving in a raw network is the construction of the C-ITS infrastructure. Similarly, in the case of the S17 | S19 link in Scenario 2, the strategy to change the road geometry is expected to bring about more advantages in terms of profitability. By applying the same method, a step-by-step construction strategy for smart infrastructure is established in specific links for each scenario.

**Table 2.** Construction strategy of smart infrastructure in Scenarios 1, 2 and 3.

| From Node | To Node | Plan | | |
|---|---|---|---|---|
| | | Scenario 1 | Scenario 2 | Scenario 3 |
| S1 | S2 | 1: constructing the C-ITS infrastructure | 1: constructing the C-ITS infrastructure | 1: constructing the C-ITS infrastructure |
| S2 | S1 | 1: constructing the C-ITS infrastructure | - | 1: constructing the C-ITS infrastructure |
| S8 | S16 | - | 1: constructing the C-ITS infrastructure | - |
| S17 | S19 | - | 2: remodeling the geometric road design | - |
| S24 | S21 | 1: constructing the C-ITS infrastructure | - | 1: constructing the C-ITS infrastructure |

Figure 6 shows the calculation results of some indices based on the smart infrastructure construction plan for each link in the optimal sub-network presented, including (a) operation gain, (b) safety gain, (c) infrastructure cost, and (d) profit.

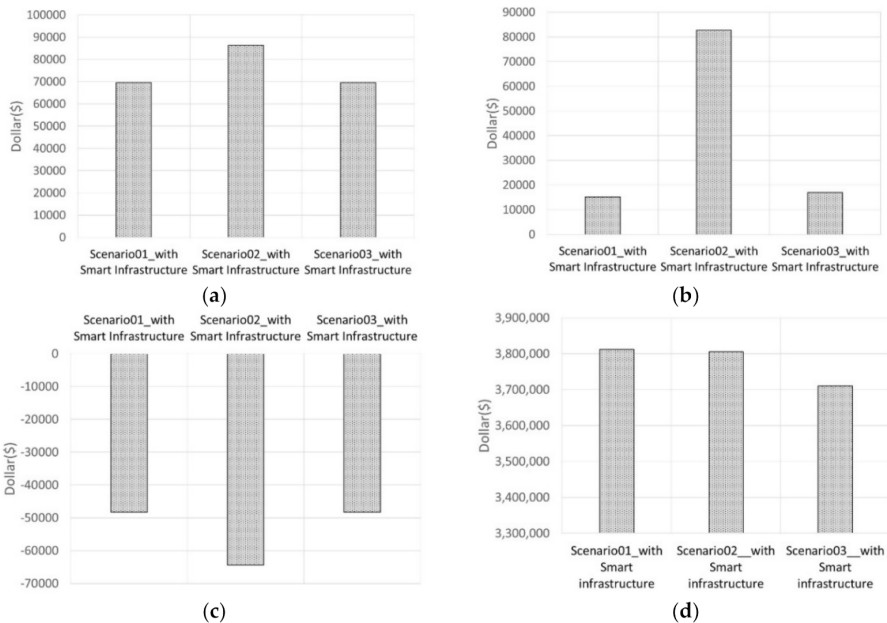

**Figure 6.** (**a**) operation gain, (**b**) safety gain, (**c**) infrastructure cost, and (**d**) total profit for optimal sub-network when constructing smart infrastructure in Scenarios 1, 2, and 3.

Figure 6a shows that there are the same operation gain in Scenarios 1 and 3 and the highest operation gain in Scenario 2. Similarly, as depicted in Figure 6b, a higher safety gain in Scenario 2 than in other scenarios can be expected which can be attributed to the difference in the construction strategy of the smart infrastructure. Unlike Scenarios 1 and 3, Scenario 2 could provide high operation and safety gains by establishing a strategy to upgrade the smart infrastructure, which adopted a specific link as a strategy to change the geometric structure corresponding to level 2. On the other hand, as shown in Figure 6c, Scenario 2 required a higher facility construction cost than the other two scenarios. The overall effect of introducing smart infrastructure is represented in Figure 6d. A higher service profitability could be achieved than in the case where smart infrastructure was not considered for the service demand pattern in every scenario. A greater improvement in profitability could be expected than in other scenarios, particularly in Scenario 2, when the CAV-M service was provided by introducing smart infrastructure.

To analyze the expected effects of introducing smart infrastructure in more detail, the results in Figure 7, which shows the result of the profit comparison between the raw network and the optimal sub-network when smart infrastructure is considered, were derived. The expected effect of introducing smart infrastructure using the difference in profit between the raw network and the optimal sub-network showed the highest utility value in Scenario 3, followed by Scenarios 2 and 1. This is somewhat different from the result of comparing profit for the CAV-M service when smart infrastructure was not considered in Figure 5. Thus, when establishing an operation plan for CAV-M service, it is necessary to introduce the CAV-M service first in scenarios where higher profitability can be achieved, considering the service demand pattern and the given autonomous driving operation conditions, rather than blindly upgrading the road network to mitigate the difficulty in applying autonomous driving.

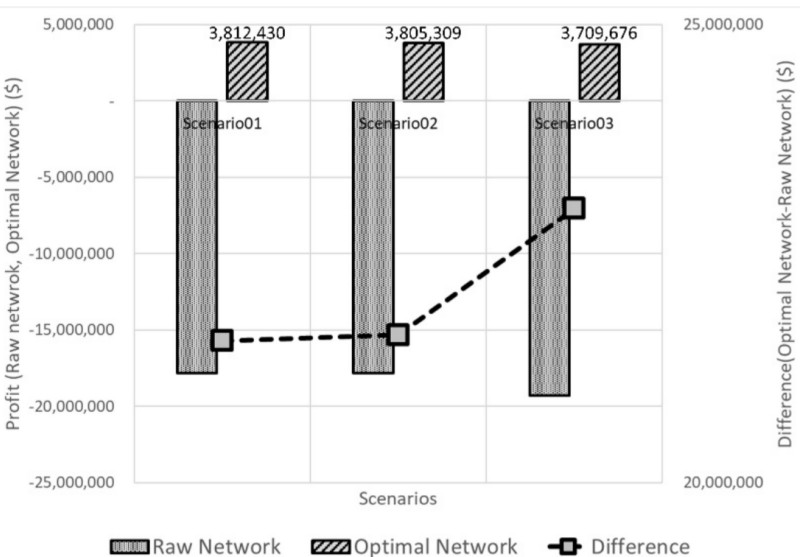

**Figure 7.** Comparison of profit between raw network and optimal sub-network when constructing smart infrastructure in Scenarios 1, 2, and 3.

Figure 8 shows the profitability distribution of the CAV-M service link-wise for the raw network and the optimal sub-network in each scenario. In Figure 8a, the majority of the links in the raw network generate insignificant but positive profits for the CAV-M service. In contrast, certain links though few generate huge expenditures that cause a significant decrease in profitability, which is the service target, leading to a devaluation of the entire network. This trend can also be seen in Scenarios 2 and 3, as shown in Figure 8b,c.

Figure 8d shows that not only the links which showed huge expenditures in the raw network generated profit, but most of the links on the service network could also generate revenue. Although some links generate expenditures, their influence appear to be insignificant, and a similar trend could be observed in the other two scenarios as well. As shown in Figure 8e, all links in the optimal sub-network in Scenario 2 made a profit and Scenario 3 can also create profits through the CAV-M service from all links, with the exception of three links.

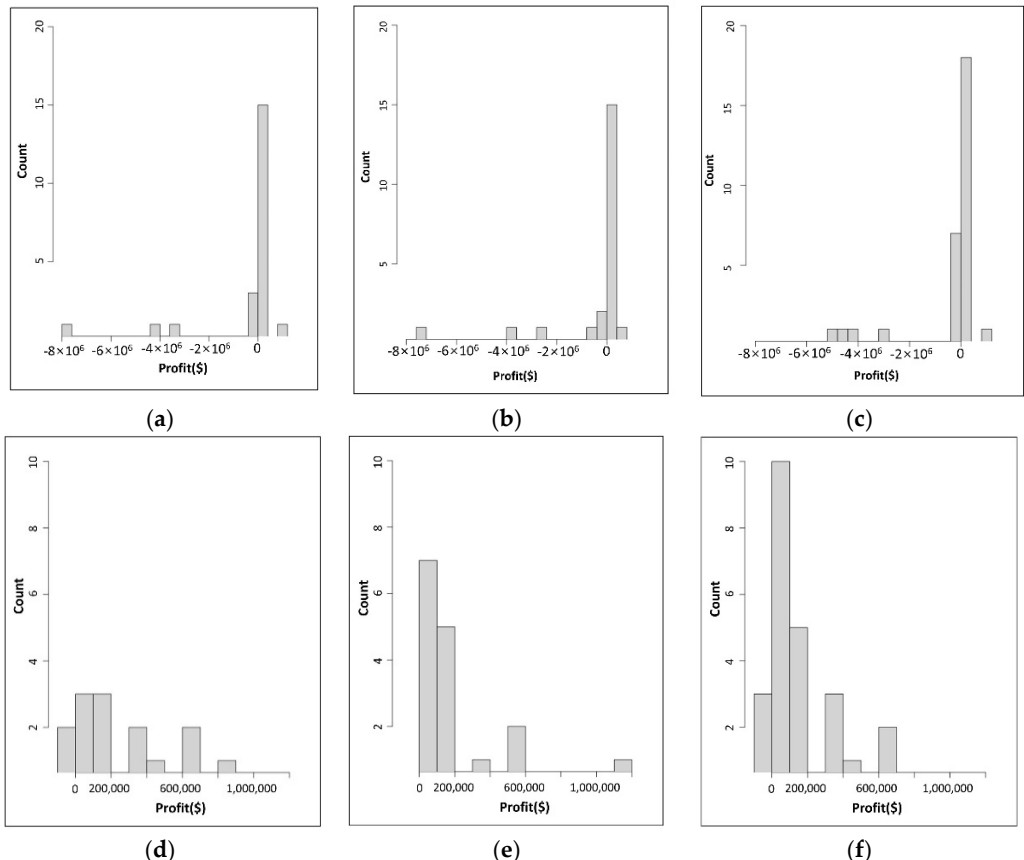

**Figure 8.** Distribution of link profit for CAV-M service in raw network and optimal sub-network for Scenarios 1, 2, and 3. (**a**) distribution of link profit for raw network in Scenario 1; (**b**) distribution of link profit for raw network in Scenario 2; (**c**) distribution of link profit for raw network in Scenario 3; (**d**) distribution of link profit for optimal sub-network in Scenario 1; (**e**) distribution of link profit for optimal sub-network in Scenario 2; (**f**) distribution of link profit for optimal sub-network in Scenario 3.

To examine the effects of the proposed methodology in more detail, the spatial distribution of each link according to the profitability of the CAV-M service is shown in Figure 9. For example, Figure 9a shows the spatial distribution of the link for the profit of the CAV-M service in the raw network for Scenario 1 where the CAV-M service is operated for 21 links in total. This service network is composed of six links that generate a service expenditure, eleven links that produce a service profit of ($)100–200,000, three links that produce a service profit of ($)200,000–400,000, and one link that produces a service profit of over ($)800,000. When it is applied to the optimal sub-network derived based on the same demand pattern of the CAV-M service, a total of 14 links are used, as shown in Figure 9d. This includes two links that generate a service expenditure, six links that produce a service profit of ($)100–200,000, two links that produce a service profit of ($)200,000–400,000, one link that produces a service profit of ($)400,000–600,000, two links that produce a service profit of ($)600,000–800,000, and one link that produces a service profit of over ($)800,000. The proposed methodology greatly contributes to the improvement of service profitability by concentrating utilization on the optimized path while intensively utilizing distributed links used to provide the CAV-M service, thus increasing the utility value in terms of the smart infrastructure facility and construction cost. This pattern can also be observed in Scenarios 2 and 3, as shown in Figure 9b,c,e,f.

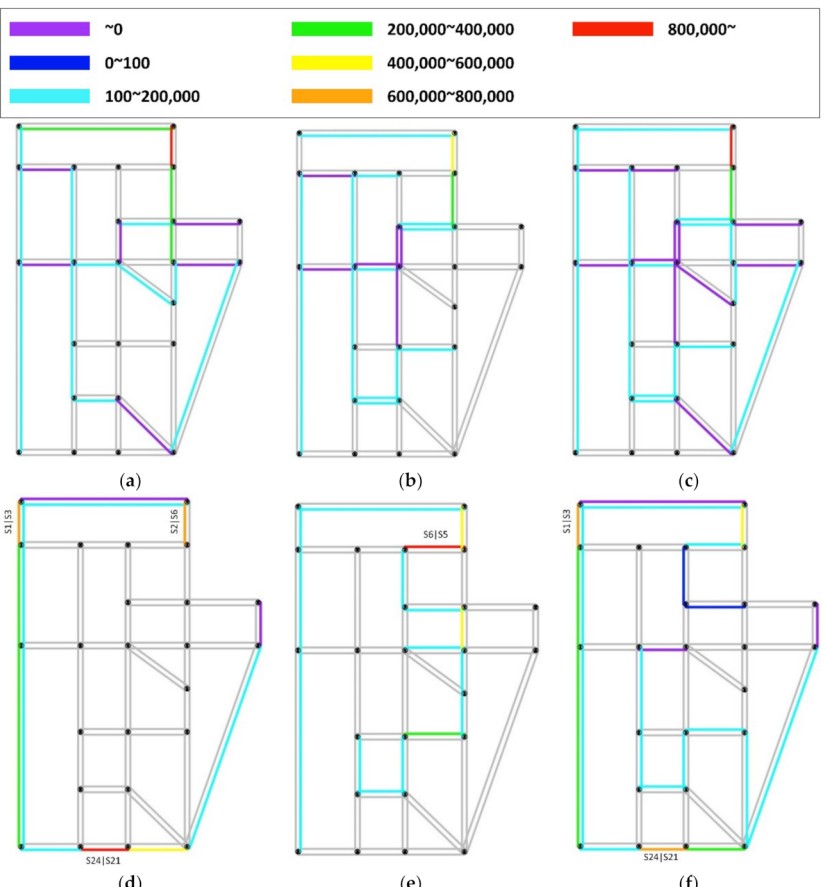

**Figure 9.** Spatial distribution of link profit for raw network and optimal sub-network in Scenarios 1, 2, and 3. (**a**) profit for each link of raw network in Scenario 1; (**b**) profit for each link of raw network in Scenario 2; (**c**) profit for each link of raw network in Scenario 3; (**d**) profit for each link of optimal sub-network in Scenario 1; (**e**) profit for each link of optimal sub-network in Scenario 2; (**f**) profit for each link of optimal sub-network in Scenario 3.

It is noteworthy that the link producing a high profit is changed by the traffic demand pattern. In the case of Figure 9d for Scenario 1, the links S24|S21, S1|S3, and S2|S6 generate high profits. In the case of Figure 9e for Scenario 2, a large profit can be expected from the link S6|S5, and in the case of Figure 9f for Scenario 3, a large profit can be expected from the S1|S3 and S24|S21 links. Thus, the proposed methodology can be used to select a road section that can guarantee the profitability of the CAV-M service by considering the characteristics of the traffic demand pattern in a specific region.

### 3.2. Service Network Analysis Varying Demand Size

In Section 3.1, we analyzed the CAV-M service network according to the change in demand pattern for the same size of traffic demand. In this section, we analyze the CAV-M service network according to the changing service demand size.

### 3.2.1. Optimal Sub-Network without Construction of Smart Infrastructure

Figure 10 shows the result of deriving the optimal sub-network in each scenario according to changing demand size when the CAV-M service is operated based on the current operation conditions without considering smart infrastructure.

Figure 10a shows the optimal sub-network in a situation where there is a large traffic demand in the network compared to the other two scenarios. In Scenario 4, the CAV-M service is provided using links corresponding to 35 among 76 links in total, whereas Scenario 5 with half the traffic demand of Scenario 4 provides the CAV-M service using

32 links as shown in Figure 10b. However, in Scenario 6 when the traffic demand decreased by 75% compared to Scenario 4, the utilization of links to support the CAV-M service decreased significantly by 31% compared to Scenario 4, as shown in Figure 10c.

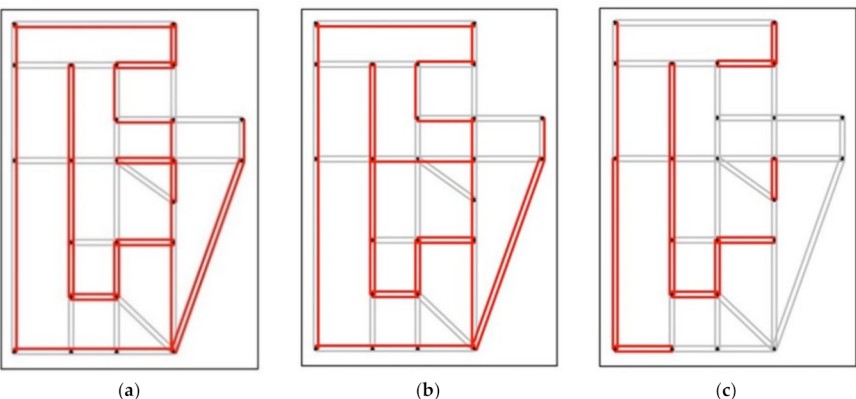

(a)  (b)  (c)

**Figure 10.** Optimal links for CAV-based mobility service in (**a**) Scenario 4, (**b**) Scenario 5, and (**c**) Scenario 6.

Figure 11 shows the demand-weighted betweenness during the provision of the CAV-M service in each scenario based on the optimal sub-network derived in Figure 10. In Figure 11a,d for Scenario 4, it can be seen through the demand-weighted betweenness score that many mobility services are supported using the links at the top (S1 | S2), leftmost links (S3 | S1, S12 | S3 and S13 | S12), and bottom links (S23 | S13, S21 | S24 and S20 | S21) in the outer part of the network. Moreover, it can be observed that the links S2 | S6, S6 | S5, S5 | S9, S9 | S8, S8 | S16, S16 | S17, S17 | S19, and S19 | S20 located in paths traversing through the network are intensively used. It can be also seen that the links S2 | S6 and S19 | S20 are actively used because they are easy to apply to automated driving owing to the relatively low operation and safety level of the links S2 | S6 and S19 | S20 compared to other links that connect the outer and inner parts of the network (e.g., links S3 | S4, S4 | S3, S12 | S1, S1 | S12, S8 | S7, S7 | S8, S24 | S23, S22 | S21 and S22 | S20). Furthermore, it can be observed that the service for internal traffic demand is supported using the links S19 | S15 and S15 | S19 with a relatively low demand weighted betweenness score.

For Scenario 5, as shown in Figure 11b,e, the overall usage frequency of the CAV-M service decreased with traffic demand. Thus, it can be observed that the demand-weighted betweenness scores of certain links that have been intensively used for movement support decreased. Moreover, in the case of Scenario 6 with a traffic demand equal to 25% that of Scenario 4, the demand-weighted betweenness score for all service links decreased by 10–4000, as shown in Figure 11c,f. Simultaneously, links that are used intensively to support the CAV-M service can be distinguished despite the sharp drop in traffic demand. Hence, the links to which the CAV-M service should be applied by prioritizing selection are clearly distinguishable.

Figure 12 shows the calculation results of the major indices for Scenarios 4, 5, and 6 that includes (a) service revenue, (b) serving ratio and increasing ratio of shortest path length, (c) operation and safety cost, and (d) profit in the optimal sub-network without considering the smart infrastructure. Figure 12a shows that the service revenue decreased in correspondence with the traffic demand in Scenarios 4, 5, and 6 thus implying that a change in the demand size has a significant impact on the service revenue. Further, Figure 12b shows the difference between the serving ratio and the increasing ratio of the shortest path length in each scenario. The serving ratio indicates the extent to which the demand is accommodated in the optimal sub-network for each scenario in Table 1. For example, a serving ratio of 1 implies that the demand is fully accommodated, whereas 0.5 implies that only 50% of the total demand is accommodated. From the results it is evident that the serving ratio of Scenarios 4 and 5 is nearly 1 while that of Scenario 6 is approximately 88%

lower than that of the other two scenarios. Owing to the relatively high traffic demands in the case of Scenario 4 and 5, economical operation seems possible by concentrating CAV-M services with specific optimal links in the optimal sub-network owing to the similarity of the OD pairs. However, in the case of Scenario 6, it seems difficult to provide economic mobility services with the optimal sub-network only because the traffic demand is evenly distributed over the entire road network and the volume of the traffic demand is low. Consequently, Scenarios 4 and 5 show a high increasing ratio of shortest path length with increasing bypasses to support a wide range of mobility services using many optimal links. Moreover, the difference in the increasing ratio of the shortest path length is not distinctive because the OD pairs are similar between the two scenarios. In contrast, Scenario 6 does not provide CAV-M services for a large part of the traffic demand; therefore, it has a low frequency of providing mobility services using a bypass and shows a relatively low increasing ratio of the shortest path length. The comparison between different scenarios in Figure 12d demonstrates that Scenario 4 has the highest profit followed by Scenarios 5 and 6: this can be attributed to the large difference in service revenue in correspondence with the volume of the traffic demand which has a direct bearing on the profit.

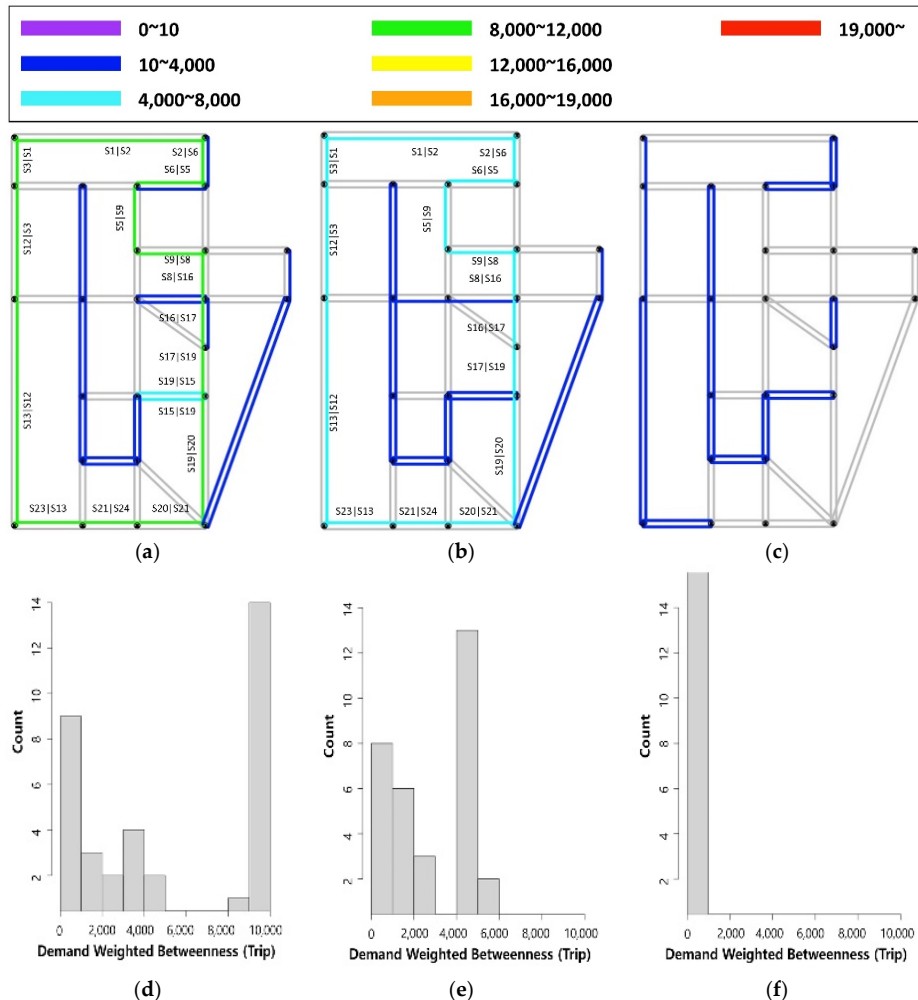

**Figure 11.** Distributions of demand weighted betweenness in Scenario 4, Scenario 5, and Scenario 6. (**a**) demand weighted betweenness in Scenario 4; (**b**) demand weighted betweenness in Scenario 5; (**c**) demand weighted betweenness in Scenario 6; (**d**) distribution of demand weighted betweenness in Scenario 4; (**e**) distribution of demand weighted betweenness in Scenario 5; (**f**) distribution of demand weighted betweenness in Scenario 6.

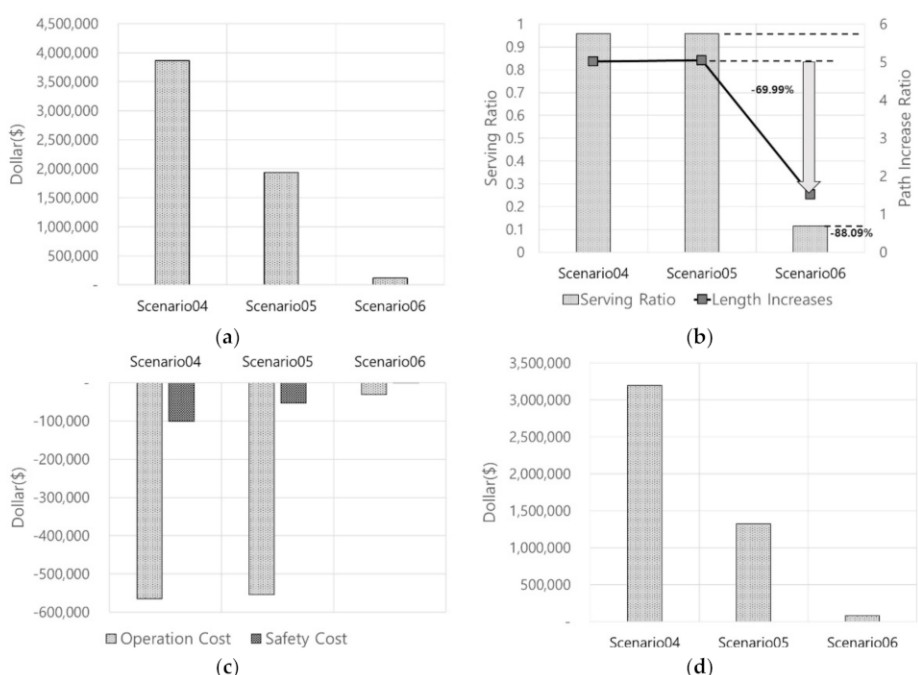

**Figure 12.** (**a**) service revenue, (**b**) increasing ratio of shortest path length, (**c**) operation and safety cost, and (**d**) total Profit in optimal sub-network without consideration of construction of smart infrastructure.

Figure 13 presents the results of comparing the raw network and the optimal sub-network for Scenarios 4, 5, and 6 without considering smart infrastructure. Figure 13a shows the differences in the service revenue and operation and safety cost between the raw network and the optimal sub-network for each scenario. Evidently, there is minor difference in the service revenue of every scenario owing to the influence of the serving ratio, introduced in Figure 12b, and the largest difference occurs in Scenario 6 which has the lowest serving ratio. Moreover, a higher difference in the operation and safety costs can be observed than in the service revenue. Particularly, because a higher difference in the safety cost is evident for all scenarios regardless of the change in the amount of traffic demand, a positive effect on the safety of the CAV-M service can be expected throughout the optimal sub-network.

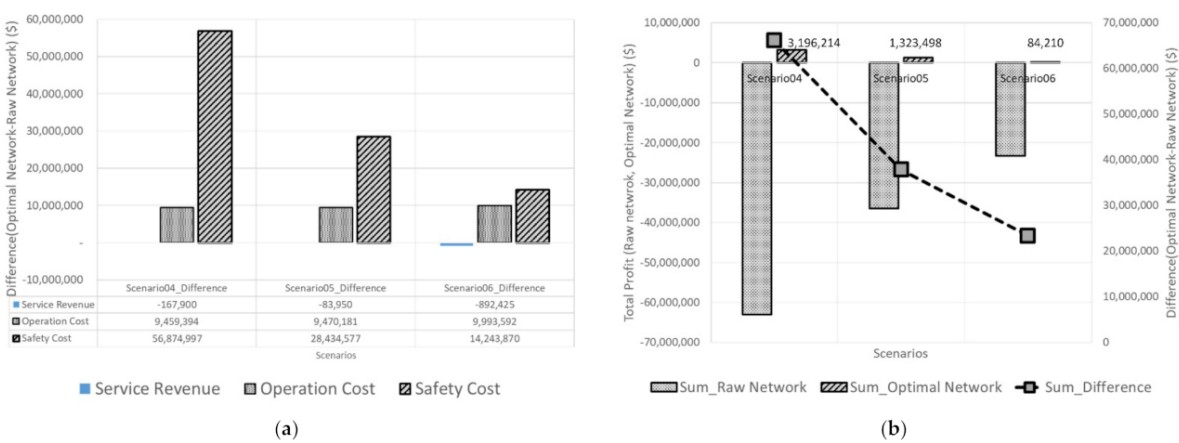

**Figure 13.** Comparison analysis of raw network, and optimal sub-network without consideration of construction of smart infrastructure. (**a**) comparision of service revenue, operation cost and safety cost between raw network and optimal network; (**b**) comparision of profit between raw network and optimal network.

As shown in Figure 13b, there is a clear difference in the profit of the CAV-M in correspondence with the volume of traffic demand. One can easily observe that the CAV-M service provided for the optimal sub-network is found to achieve profitability in every scenario regardless of the size of traffic demand. Particularly when the volume of traffic demand is high, the difference in profit between the raw network and the optimal sub-network is large. Furthermore, it is evident that the importance of sub-network selection for service provision has increased with the demand for autonomous cooperative driving mobility in line with this trend.

### 3.2.2. Optimal Sub-Network with Construction of Smart Infrastructure

Table 3 shows the construction strategy of smart infrastructure for the optimal sub-network in Scenarios 4, 5, and 6 derived through the GA-based optimization described in Section 2.

**Table 3.** Construction strategy of smart infrastructure in Scenarios 4, 5 and 6.

| From Node | To Node | Plan | | |
| --- | --- | --- | --- | --- |
| | | Scenario 1 | Scenario 2 | Scenario 3 |
| S1 | S2 | 1: constructing the C-ITS infrastructure | 1: constructing the C-ITS infrastructure | - |
| S17 | S19 | 1: constructing the C-ITS infrastructure | 2: remodeling the geometric road design | - |
| S19 | S20 | 1: constructing the C-ITS infrastructure | 1: constructing the C-ITS infrastructure | - |

As shown in Table 3, step-by-step construction strategies for smart infrastructure are established for specific links in each scenario. Particularly for Scenario 6, it is expected that not investing in the construction of smart infrastructure will bring benefits in terms of profitability for links that have a low demand for the CAV-M service because they are distributed sporadically in the optimal sub-network.

Figure 14 shows the calculation results of the major indices derived according to the smart infrastructure construction plan for each link in the optimal sub-network, which includes (a) operation gain, (b) safety gain, (c) infrastructure cost, and (d) profit.

Figure 14a shows that an operation gain could be attained in Scenarios 4 and 5 owing to relatively high demand, whereas operation gain could not be attained in Scenario 6 because of a low demand resulting from non-installation of any smart infrastructure facility.

As shown in Figure 14b, it can be seen that akin to the result for operation gain no safety gain is generated in Scenario 6, whereas a significant difference between Scenarios 4 and 5 can be observed, unlike the operation gain. Consequently, a high safety gain can be expected from the installation of smart infrastructure in Scenario 4 with a relatively large traffic demand and a high degree of concentration for specific links. Considering that the cost for constructing a smart infrastructure facility in Scenarios 4 and 5 is the same, as indicated in Figure 14c, it can be said that Scenario 4 has a better probability for a greater return on the investment.

As depicted in Figure 14d, the highest profit can be expected in Scenario 4, followed by Scenarios 5 and 6: a trend similar to the result shown in Figure 12d described above. The two scenarios, excluding Scenario 6, could achieve a slightly higher profit than the case of not considering smart infrastructure, thus implying that constructing smart infrastructure facilities in the right place according to service demand could effectively increase the profitability of the CAV-M service.

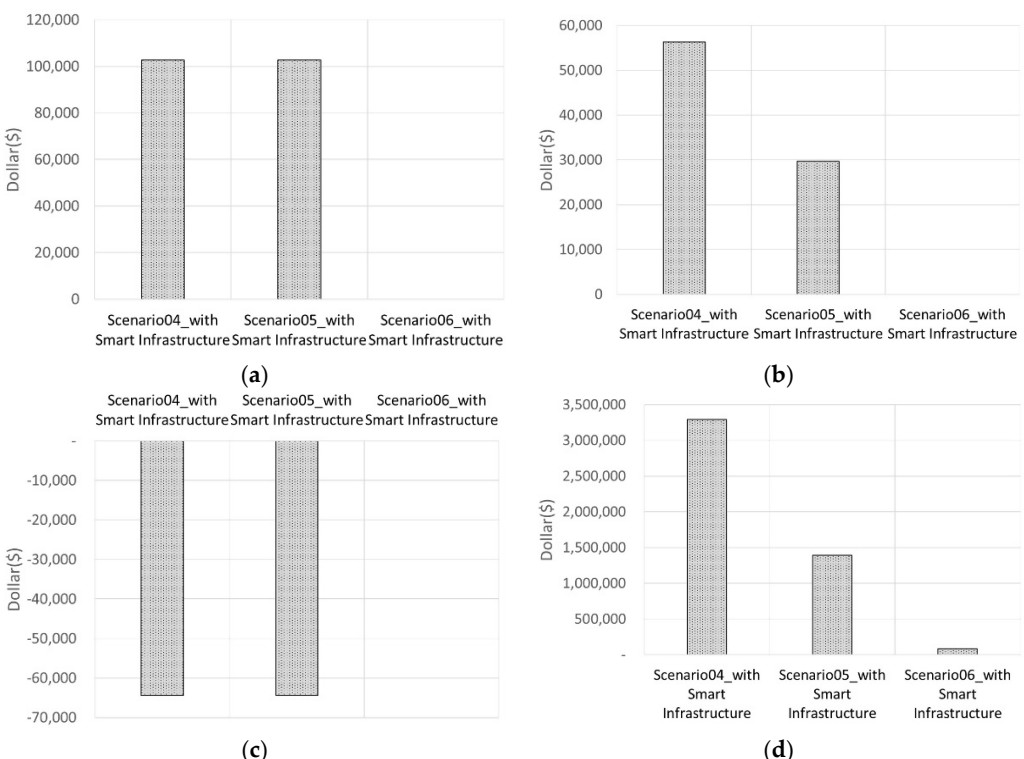

**Figure 14.** (**a**) operation gain, (**b**) safety gain, (**c**) infrastructure cost, and (**d**) total profit when constructing smart infrastructure in Scenarios 4, 5, and 6.

The detailed results of the expected effects of introducing smart infrastructure are shown in Figure 15.

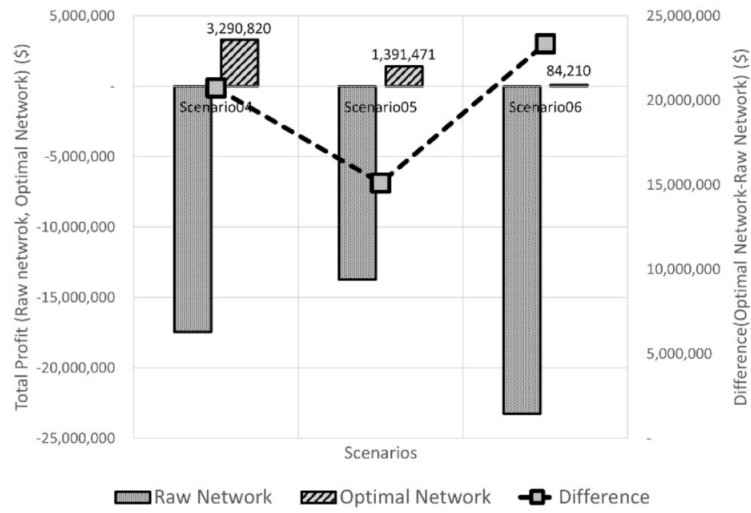

**Figure 15.** Comparison analysis of raw network and optimal sub-network when constructing smart infrastructure in Scenarios 4, 5, and 6.

As shown in Figure 15, the profit difference between the optimal sub-network and raw network in correspondence with decreasing demand is different under conditions when the smart infrastructure is installed and when it is not installed. A profit of ($)84,210 in the optimal sub-network and an expenditure of ($)23,260,827 in the raw network in Scenario 6 could be observed. It is worth noting that no change in profit could be evidenced because the smart infrastructure was not installed. Despite the positive effect of smart infrastructure, it is difficult to expect profit creation through the CAV-M service in the raw network, and it

was found that the process of deriving the optimal sub-network was required when the traffic demand was large and the installation of smart infrastructure was also considered.

Figure 16 shows the distribution of profits of the CAV-M service by link for the raw network and the optimal sub-network in Scenarios 4, 5, and 6. In Figure 16a, which shows the profit distribution of the raw network corresponding to Scenario 4, some links exhibit a meagre positive profit from the CAV-M service. However, most links show expenditures below ($)0.5M while a few links generate huge expenditures indicating a decrease in the utility of the CAV-M service for the entire network. This trend is also observed in Scenarios 5 and 6, as shown in Figure 8b,c. Figure 16d shows that all links, with the exception of one, generated a profit. Besides, some links that generated large expenditures in the raw network generated profits: a similar trend can also be observed in the other two scenarios. As shown in Figure 16e, all except four links in the optimal sub-network switched to make a profit in Scenario 5. Likewise, all links except one showed a profit from the CAV-M service in Scenario 6, as shown in Figure 16f. These results provide sufficient evidence for the superiority of the proposed method on the CAV-M service introduction strategy while considering changes in the volume of traffic demand, automated driving operation conditions, and road driving environment.

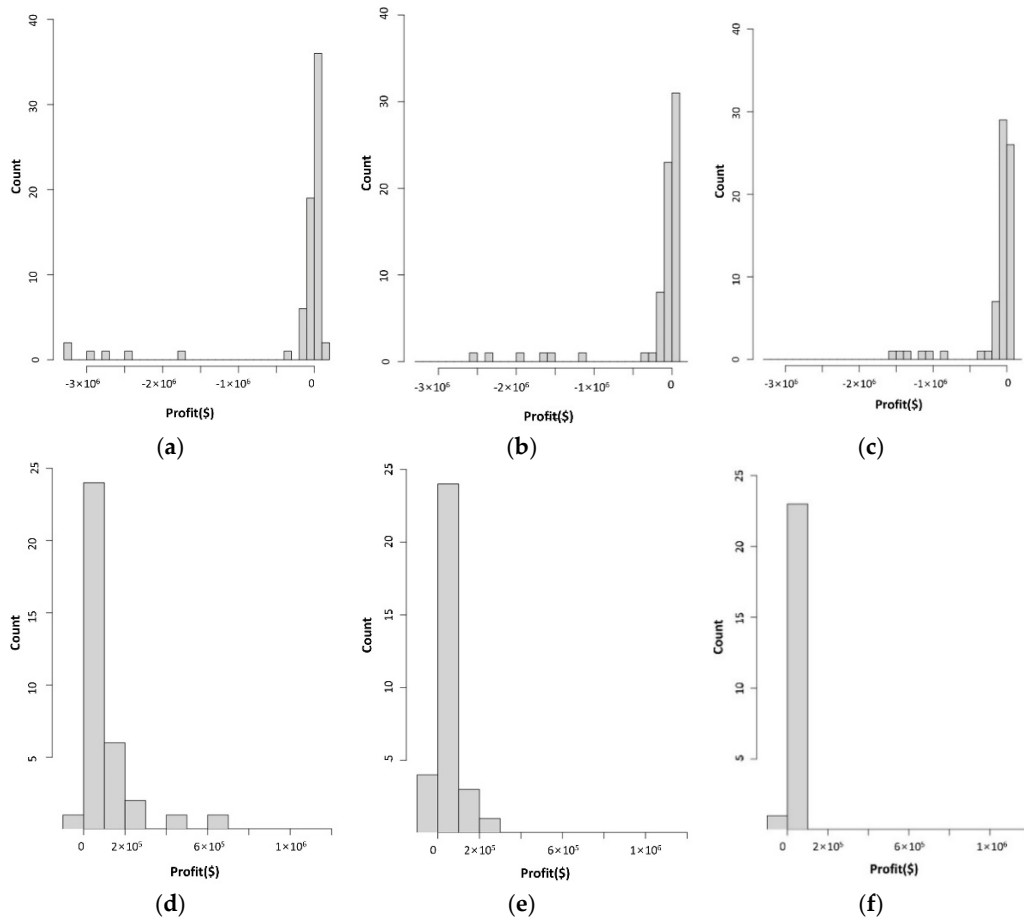

**Figure 16.** Distribution of link profit for CAV-M service in raw network and optimal sub-network for Scenarios 4, 5, and 6. (**a**) distribution of link profit for raw network in Scenario 4; (**b**) distribution of link profit for raw network in Scenario 5; (**c**) distribution of link profit for raw network in Scenario 6; (**d**) distribution of link profit for optimal sub-network in Scenario 4; (**e**) distribution of link profit for optimal sub-network in Scenario 5; (**f**) distribution of link profit for optimal sub-network in Scenario 6.

To analyze the proposed method in detail, the spatial distribution of each link according to the profitability of the CAV-M service is shown in Figure 17. It can be observed

from the spatial distribution for each link of the CAV-M service profit in the raw network, shown in Figure 17a, that the CAV-M service is provided using 70 links in total. This service network is composed of 32 links that generate service expenditure, 38 links that produce a service profit of ($)100–200,000. When the traffic demand volume is equal and is applied to the optimal sub-network in charge of the CAV-M service, a total of 35 links are used, as shown in Figure 17d. This network includes one link that generates a service expenditure, 29 links that produce a service profit of ($)100–200,000, four links that produce a service profit of ($)200,000–400,000, and one link that produces a service profit of ($)600,000–800,000.

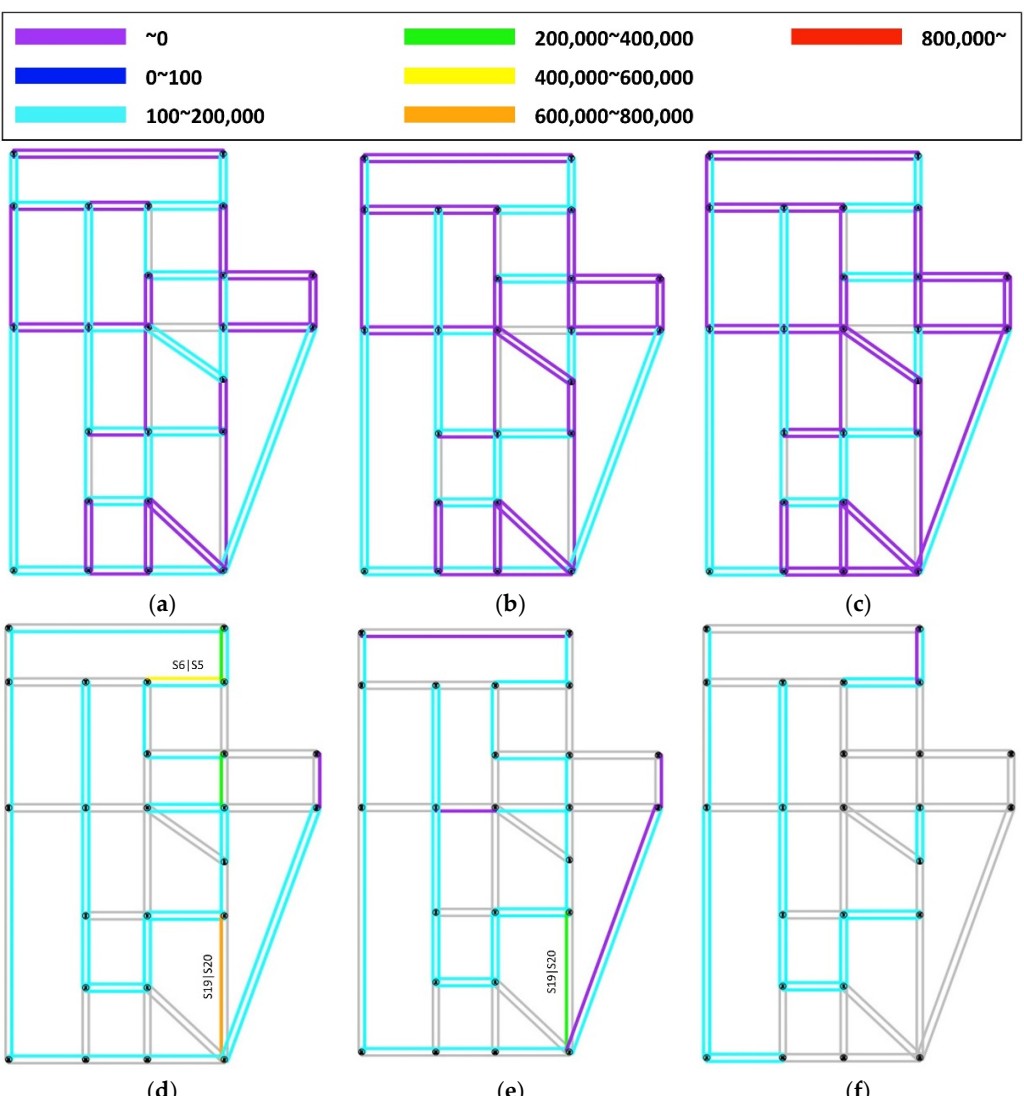

**Figure 17.** Spatial distribution of link profit for raw network and optimal sub-network in Scenarios 4, 5, and 6. (**a**) profit for each link of raw network in Scenario 4; (**b**) profit for each link of raw network in Scenario 5; (**c**) profit for each link of raw network in Scenario 6; (**d**) profit for each link of optimal sub-network in Scenario 4; (**e**) profit for each link of optimal sub-network in Scenario 5; (**f**) profit for each link of optimal sub-network in Scenario 6.

It is noteworthy that the existence or absence of links producing a high profit is determined by the volume of the traffic demand. In Figure 17d for Scenario 4, the links S19 | S20 and S6 | S5 produced a relatively high profit, whereas in Figure 17e for Scenario 5, the link S6 | S5 produced a relatively low profit. Furthermore, in Figure 17f for Scenario 6, there is no link that produces a high profit. Thus, it is likely that a major CAV-M service

link that is selected when the traffic demand is high may not be a link that guarantees profitability when the traffic demand is low.

## 4. Conclusions

This study aimed to develop an optimization method for the introduction and operation of CAV-M services in an automated driving environment. Hence, this study proposed an optimal sub-network extraction method that can maximize the service operation profit at the road network level. The proposed method extracts a sub-network that can maximize the service operation profit by performing GA optimization based on cost modeling in consideration of the traffic demand by road section, difficulty in applying automated driving, and smart infrastructure operation level for the target area of CAV-M.

This study analyzed the effectiveness of the proposed method according to various demand changes based on the Sioux Falls network, which is widely used in the transportation field. To analyze the characteristics of the optimal sub-network of the service area derived through the proposed method, the importance of each road section was assessed using the demand-weighted betweenness derived from graph theory. It was observed that the relative importance of road sections changed within the optimal sub-network according to changes in traffic demand and the amount of demand in the service area. This allowed identification of the links that should be considered first when introducing the CAV-M service.

This research examined changes in revenue by item according to changing demand in the raw network and the optimal sub-network of the service area. We observed that there were road sections that need to be serviced first in order to achieve a greater economic effect and that the CAV-M service should be applied taking the difficulty in applying automated driving into consideration. Furthermore, even though operating the CAV-M service in the raw network generated a huge expenditure, it was found that profitability can be achieved for every demand pattern when the CAV-M service is operated using the optimal sub-network derived by the proposed method. It is evidence to prove the excellence of the proposed method.

The change in the profit of the CAV-M service was analyzed through step-by-step construction strategies of smart infrastructure installed to mitigate the difficulty in applying automated driving. We found that the expected effect of introducing smart infrastructure using the difference in profit between the raw network and the optimal sub-network was somewhat different from the result of comparing the CAV-M service profits when smart infrastructure was not considered. This is because the positive influence of smart infrastructure is more profound in the raw network than in the optimal sub-network. Nevertheless, it is difficult to achieve profits through CAV-M services in the raw network. In conclusion, providing the CAV-M service using the optimal sub-network can be a better choice in terms of profitability when the installation of smart infrastructure is considered together in a situation with a large traffic demand.

This study analyzed the profitability of CAV-M services from the perspective of road operation, considering changes in traffic demand and driving limitations of automated driving vehicles. However, several issues still need to be further considered in future study. Sensitivity analyses with respect to hyperparameter values related to the unit costs of the proposed model should be conducted in near future since the specified values of the parameter involved in the unit costs have significant impact on extracting the optimal sub-network for the CAV-M service. In this study, the optimal sub-network was derived through optimization based on GA. There are still additional research tasks to be performed to conduct sensitivity analysis of the proposed model with respect to the parameters involved in the population, crossover, mutation and elitism of the GA. Moreover, additional analysis on the effectiveness of the optimal sub-network by applying other optimization methods based on metaheuristics such as particle swarm optimization or ant colony optimization is also considered in future study. Furthermore, it is necessary to examine changes in service revenue when different methods of providing a path that affects the CAV-M serving

ratio are used. In addition, it would be necessary to develop a road classification system that reflects the driving limitations of automated driving vehicles based on the analysis of real-road automated driving conditions in future research. Based on this research, the ripple effects of more detailed CAV-M services could be investigated by adjusting the profit model based on the safety and difficulty of automated driving according to the road environment type. Furthermore, a study on the method of constructing further improved mobility systems and mobility services in consideration of a public transportation strategy linked to the CAV-M service will be necessary. Research work for efficient utilization of road networks through research on the method of connecting with future mobility services, such as shared mobility, will also be needed.

**Author Contributions:** S.T.: Conceptualization, Data curation, Formal analysis, Funding acquisition, Methodology, Software, Visualization, Writing—Original draft; J.K.: Methodology, Investigation, Writing—Original draft; D.L.: Formal analysis, Validation, Writing—Original draft, Writing—Review & editing. All authors have read and agreed to the published version of the manuscript.

**Funding:** This research was supported by the National Research Council for Economics, Humanities and Social Sciences and the Korea Transport Institute under the Strategy of Digitalization of Road Network for Connected and Automated Driving Research Project (Grant No. 21-22-014).

**Institutional Review Board Statement:** Not applicable.

**Informed Consent Statement:** Not applicable.

**Data Availability Statement:** Not applicable.

**Conflicts of Interest:** The authors declare no conflict of interest.

## Appendix A

The Sioux Falls network and link characteristics such as capacity, length, and speed limits have been generated by referring to [39]. The Sioux Falls network consists of 24 nodes and 76 directed links, as shown in Figure A1: Table A1 summarizes the link characteristics of the network. Each link has five attributes: from-node, to-node, length of links, operation and safety level, and infrastructure level. The from-node, to-node, and length of links are used as given in the datasets. The link length is used to calculate the path lengths and to find the shortest path. The attributes of operation and safety level are generated randomly among 1–5 levels. The operation and safety levels are used to calculate $C_{Cost}^{Service}$ as $C_{Cost}^{Operation\ Level}$, $P_{Safety}^{Collision}$, and $C_{Safety}^{Severity}$. In real-world applications, this attribute needs to be evaluated based on the conditions of road environments. The attribute of the infrastructure level is given as 0 because it is assumed that there is no smart infrastructure in the network. The value of the infrastructure level of each link is determined after solving the optimization problem stated above.

**Table A1.** Link properties for network analysis from CAV perspective.

| From Node | To Node | Length | Operation and Safety Level | Infrastructure Level (before) | From Node | To Node | Length | Operation and Safety Level | Infrastructure Level (before) |
|---|---|---|---|---|---|---|---|---|---|
| S1 | S2 | 0.4 | 3 | 0 | S13 | S24 | 0.9 | 1 | 0 |
| S1 | S3 | 0.3 | 1 | 0 | S14 | S11 | 0.3 | 1 | 0 |
| S2 | S1 | 1 | 2 | 0 | S14 | S15 | 0.5 | 2 | 0 |
| S2 | S6 | 0.2 | 1 | 0 | S14 | S23 | 2.9 | 1 | 0 |
| S3 | S1 | 3.2 | 1 | 0 | S15 | S10 | 0.6 | 4 | 0 |
| S3 | S4 | 0.2 | 3 | 0 | S15 | S14 | 0.5 | 2 | 0 |
| S3 | S12 | 0.1 | 2 | 0 | S15 | S19 | 1.2 | 1 | 0 |
| S4 | S3 | 0.4 | 4 | 0 | S15 | S22 | 1.3 | 1 | 0 |
| S4 | S5 | 0.4 | 4 | 0 | S16 | S8 | 1.2 | 2 | 0 |
| S4 | S11 | 0.2 | 1 | 0 | S16 | S10 | 1.6 | 2 | 0 |
| S5 | S4 | 1.7 | 3 | 0 | S16 | S17 | 0.2 | 1 | 0 |
| S5 | S6 | 1.7 | 1 | 0 | S16 | S18 | 0.6 | 5 | 0 |
| S5 | S9 | 3.2 | 2 | 0 | S17 | S10 | 0.6 | 3 | 0 |

**Table A1.** *Cont.*

| From Node | To Node | Length | Operation and Safety Level | Infrastructure Level (before) | From Node | To Node | Length | Operation and Safety Level | Infrastructure Level (before) |
|---|---|---|---|---|---|---|---|---|---|
| S6 | S2 | 1.7 | 1 | 0 | S17 | S16 | 0.2 | 1 | 0 |
| S6 | S5 | 0.3 | 1 | 0 | S17 | S19 | 1.5 | 2 | 0 |
| S6 | S8 | 0.9 | 2 | 0 | S18 | S7 | 2.9 | 3 | 0 |
| S7 | S8 | 0.5 | 4 | 0 | S18 | S16 | 0.6 | 5 | 0 |
| S7 | S18 | 0.2 | 2 | 0 | S18 | S20 | 0.3 | 2 | 0 |
| S8 | S6 | 1.7 | 3 | 0 | S19 | S15 | 1 | 1 | 0 |
| S8 | S7 | 3.2 | 3 | 0 | S19 | S17 | 0.7 | 3 | 0 |
| S8 | S9 | 0.3 | 2 | 0 | S19 | S20 | 0.9 | 2 | 0 |
| S8 | S16 | 1.1 | 2 | 0 | S20 | S18 | 0.5 | 2 | 0 |
| S9 | S5 | 0.3 | 2 | 0 | S20 | S19 | 2.9 | 2 | 0 |
| S9 | S8 | 0.2 | 2 | 0 | S20 | S21 | 0.9 | 2 | 0 |
| S9 | S10 | 0.5 | 5 | 0 | S20 | S22 | 1.6 | 3 | 0 |
| S10 | S9 | 0.5 | 5 | 0 | S21 | S20 | 0.3 | 2 | 0 |
| S10 | S11 | 0.2 | 3 | 0 | S21 | S22 | 0.5 | 3 | 0 |
| S10 | S15 | 3.2 | 3 | 0 | S21 | S24 | 1.6 | 2 | 0 |
| S10 | S16 | 1.3 | 2 | 0 | S22 | S15 | 0.4 | 1 | 0 |
| S10 | S17 | 0.5 | 2 | 0 | S22 | S20 | 0.7 | 4 | 0 |
| S11 | S4 | 0.2 | 1 | 0 | S22 | S21 | 0.5 | 4 | 0 |
| S11 | S10 | 0.6 | 4 | 0 | S22 | S23 | 0.3 | 1 | 0 |
| S11 | S12 | 0.6 | 5 | 0 | S23 | S14 | 0.7 | 1 | 0 |
| S11 | S14 | 0.3 | 1 | 0 | S23 | S22 | 0.3 | 1 | 0 |
| S12 | S3 | 1.1 | 2 | 0 | S23 | S24 | 1.2 | 3 | 0 |
| S12 | S11 | 0.6 | 5 | 0 | S24 | S13 | 0.3 | 1 | 0 |
| S12 | S13 | 0.3 | 1 | 0 | S24 | S21 | 0.4 | 2 | 0 |
| S13 | S12 | 1.9 | 1 | 0 | S24 | S23 | 0.4 | 4 | 0 |

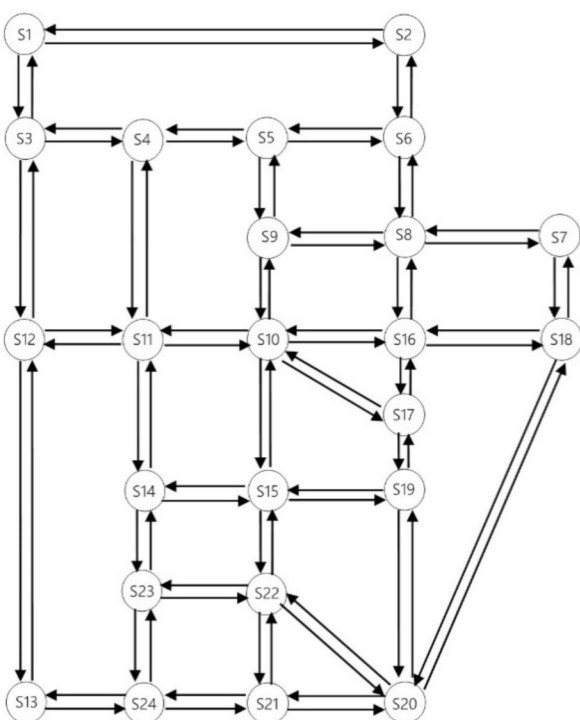

**Figure A1.** Simplified Sioux Falls network.

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
