# Peer review of "Study on the Extraction Method of Sub-Network for Optimal Operation of Connected and Automated Vehicle-Based Mobility Service and Its Implication"

_sustainability, doi:10.3390/su14063688_

Round 1
Reviewer 1 Report
This paper proposes a bi-level optimization to first identify the optimal road network and then the optimal operation and construction strategy, with the purpose of maximizing the total profit for imposing the CAV-M services. The underlying problem is important. However, the current manuscript requires extensive modification to be acceptable:
Major issues:
1. The draft is unnecessarily long.
2. There are so many hyper-parameters in e.g., eqs (5), (7), (8), (11), (12). The robustness of the conclusion w.r.t. those hyper-parameters are unknown.
3. GA can only guarantee local optimality, instead of global optimality. Moreover, the local optimum depends on the initial conditions. No sensitivity analysis w.r.t the initial condition is given.
Minor issues:
1. Eq (2), how to determine C^{service}_{Revenue}.
2. Suggest adding node numbers in Fig. 2.
3. The resolution of all the figures is very low. Please redraw them with at least 300 dpi.
Reviewer 2 Report
Journal: Sustainability (ISSN 2071-1050)
Manuscript ID: sustainability-1632449
Title: Study on the Extraction Method of Sub-network for Optimal Operation of Connected and Automated Vehicle-based Mobility Service and Its Implication
A) General Considerations
The work entitled Study on the Extraction Method of Sub-network for Optimal Operation of Connected and Automated Vehicle-based Mobility Service and Its Implication develops a framework for maximizing the profit of connected and automated vehicle-based mobility (CAV-M) service using cost modeling and metaheuristic optimization algorithm. The work is interesting and especially useful for a public more involved with the subject. The document is well written, in clear language. The structure of the document is adequate, and the conclusions of the work are well aligned with the objectives initially outlined. The work is divided into 4 main sections: (1) Introduction; (2) Methodology (Sub-network extraction method, Network and scenarios); (3) Results (Service network analysis varying demand patterns, Service network analysis varying demand size); and (4) Conclusions.
(1) Introduction
In this section, the authors make a very interesting presentation about their motivation for approaching this topic. The authors remember that the main objective of this research is to develop a framework for maximizing the profit of connected and automated vehicle-based mobility (CAV-M) service. The authors propose a framework that selects the optimal links from the road network for the CAV-M service to maximize the profit, where the service area is limited by the ODD. To solve this specific problem the authors propose a methodology to evaluate the profit of each road link and sub-network based on graph theory. Also, the authors propose an integrated methodology to extract an optimal sub-network for CAV-M services and to establish a contraction strategy for smart infrastructure by using a meta-heuristic algorithm.
(2) Methodology
In this section, the authors present the framework of this study and explain the detailed methodology for each analysis: the sub-network extraction method (link performance evaluation method, and optimization with genetic algorithm), and the network and scenarios. The authors emphasize the optimization framework to find optimal operation and construction strategy used and underline the demand scenarios constructed.
(3) Results
This section deals with the comparative analysis of the raw network and optimal sub-network varying demand pattern and demand amount. Thus, authors go deep detailing either the Service network analysis varying demand patterns (optimal sub-network without - and with - construction of smart infrastructures), or the Service network analysis varying demand size (optimal sub-network without - and with - construction of smart infrastructures too). This section is particularly enriched by several data, details, and discussions concerning: Optimal links for CAV-M service with varying demand patterns; Demand weighted betweenness Scenarios; Comparison analysis of raw network, and optimal sub-network without consideration of the construction of smart infrastructure; Comparison analysis of raw network and optimal sub-network when constructing smart infrastructure; Comparison of profit between raw network and optimal sub-network when constructing smart infrastructure; Distribution of link profit for CAV-M service; Optimal links for CAV-based mobility service; Spatial distribution of link profit for raw network and optimal sub-network; among others.
(4) Conclusions
In this section, the authors summarize the results and suggest directions for future research. In fact, the authors use this section to make a very interesting and in-depth summary of the work carried out: the motivation for the work and the objectives initially proposed; the methodology adopted to achieve the results; and the results obtained and the discussion around them are reviewed. Likewise, it is worth highlighting the future research tasks that the authors point out for the continuation of this work, among which we emphasize: to apply other optimization methods based on metaheuristics such as particle swarm optimization or ant colony optimization; and to examine changes in service revenue when different methods of providing a path that affects the CAV-M serving ratio are used.
Round 2
Reviewer 1 Report
The manuscript in the current form is acceptable.